# TOWARDS CONTROL-CENTRIC REPRESENTATIONS IN REINFORCEMENT LEARNING FROM IMAGES

## ABSTRACT

Image-based Reinforcement Learning is a practical yet challenging task. A major hurdle lies in extracting control-centric representations while disregarding irrelevant information. While approaches that follow the bisimulation principle exhibit the potential in learning state representations to address this issue, they still grapple with the limited expressive capacity of latent dynamics and the inadaptability to sparse reward environments. To address these limitations, we introduce ReBis, which aims to capture control-centric information by integrating reward-free control information alongside reward-specific knowledge. ReBis utilizes a transformer architecture to implicitly model the dynamics and incorporates block-wise masking to eliminate spatiotemporal redundancy. Moreover, ReBis combines bisimulation-based loss with asymmetric reconstruction loss to prevent feature collapse in environments with sparse rewards. Empirical studies on two large benchmarks, including Atari games and DeepMind Control Suit, demonstrate that ReBis has superior performance compared to existing methods, proving its effectiveness.

## 1 INTRODUCTION

Practical applications of reinforcement learning (RL) necessitate the ability to teach an agent to control itself in an environment with a visually intricate observation space. When visual signals serve as observations, the agent continuously receives images during its interaction with the environment. These images are not only temporally correlated but also carry substantial spatial redundancy (He et al., 2022; Feichtenhofer et al., 2022; Bao et al., 2022), which can potentially introduce distractions and noise to prevent the agent from yielding desired policies. Imagine an RL agent with the goal of seeking a target position while being confronted with a TV emitting uncontrollable random noise. The observation of noisy TV should not distract the agent's attention to finding its path as it is neither relevant to the agent's control nor helpful in getting a higher reward. In such a scenario, it is imperative for the agent to learn the representation of the environment that captures relevant information for control while ignoring irrelevant information.

Representation learning tailored for RL is a promising way to improve the perception of the agent by extracting information from noisy observations into low-dimensional vectors. Common approaches include reconstructing observations via an autoencoder (Yarats et al., 2021c), applying data augmentation (Yarats et al., 2021b; Laskin et al., 2020), or devising auxiliary tasks (Yu et al., 2022; 2021; Fedus et al., 2019; Jaderberg et al., 2017) to reduce redundancies in observation. However, they cannot guarantee the preservation of task-specific information in decision-making tasks. Behavioral metrics (Liao et al., 2023; Chen & Pan, 2022) appear to be a promising solution to mitigate this issue.

A prominent category of behavioral metrics, named Bisimulation metrics (Ferns et al., 2004; 2006; Castro, 2020), aims to capture structures in the environment by learning a metric that measures behavioral similarities between states. This behavioral similarity considers the distance between (i) their immediate rewards and (ii) their transition distributions, thereby guiding the agent's focus toward the task it is supposed to solve. Recent work (Zhang et al., 2021b; Castro et al., 2021; Zang et al., 2022) has successfully applied the bisimulation principle to shape the representations of deep RL agents to capture task-specific information and accelerate policy learning.

However, we have identified that there still are theoretical obstacles when applying bisimulation-based approaches in practical state representation learning. Firstly, the convergence of bisimulation metrics requires an unbiased estimation when incorporating latent dynamics modeling. However, modeling

via Gaussian distribution is notably restricted, especially when the underlying distribution is multi-modal. This limitation results in substantial approximation error, which might inadvertently disrupt the representation learning process. Secondly, control-relevant but reward-free information could be vital in environments with uninformative rewards, such as sparse or near-constant rewards. In these cases, bisimulation objectives might inaccurately assume all states to be equivalent, leading to collapsed representations. Therefore, at the control level, a model capable of forecasting spatiotemporal information may produce informative representations that are beneficial for dynamic modeling and effective in guiding the agent's actions. This highlights the importance of not only guaranteeing bisimulation capability but also mitigating spatio-temporal redundancy. Achieving this balance is essential for empowering the agent with the ability to understand and learn control-centric information.

We propose ReBis (latent REconstruction with BISimulation measurement), to learn control-centric representations and effectively address the aforementioned issues. Intuitively, reconstructing a visual signal with high information density through low-dimensional feature embeddings, a common practice in the computer vision domain, can successfully preserve spatiotemporal information. However, it is unnecessary and inefficient to reconstruct at the pixel level as this contains significant redundancies. Therefore, we opt to reconstruct the latent features instead of raw observations, maintaining essential information relevant to control while reducing unnecessary spatiotemporal redundancies.

Considering that an appropriate representation should be expressive enough to encapsulate the dynamics while remaining practically tractable, we start with bisimulation objectives with approximate dynamics. Consequently, we utilize the transformer architecture (Vaswani et al., 2017) to implicitly model the forward dynamics, thereby enhancing the awareness of multi-modal behavior while extracting temporal information from the observation sequences. To reduce spatiotemporal redundancy and better capture the reward-free control information, we incorporate Block-wise masking (Wei et al., 2022) to minimize the interference of irrelevant exogenous spatiotemporal noise in the observation space. Moreover, we address the challenge of bisimulation objectives collapsing in environments with sparse rewards by developing an asymmetric latent reconstruction loss that effectively prevents failure cases, ensuring the soundness of our model. ReBis serves as a state representation learning module and can be seamlessly integrated into any existing downstream RL framework to enhance the agent's understanding of the environment. We summarize our main contributions as follows:

- We recognize the limitations of previous work that adheres to the bisimulation principle for RL representation learning. Our study highlights the importance of a highly expressive dynamics model and the necessity of capturing reward-free control information.
- We propose ReBis as an efficient method for learning state representation tailored to vision-based RL, encompassing both spatial-temporal consistency and long-term behavior similarity.
- We demonstrate the superior performance of ReBis on large benchmarks, including Atari games (Bellemare et al., 2013) and DeepMind Control Suite (Tassa et al., 2018).

## 2 PRELIMINARIES

### 2.1 IMAGE-BASED RL

We begin by introducing the notations and outlining the realistic assumptions regarding the underlying structure in the environment, as the paper focuses on the image-based RL tasks.

In most practical settings, the agent does not have access to the actual states while interacting with the environment. Instead, it receives limited information through observations (Bharadhwaj et al.). We consider the learning process as a partially observable Markov decision process (POMDP), which is formulated as $(\mathcal{O}, \mathcal{S}, \mathcal{A}, P, \rho, r, \gamma)$, including a potentially infinite observation space $\mathcal{O}$ (*e.g.*, pixels), a low-dimensional latent state space $\mathcal{S}$, and an action space $\mathcal{A}$. The latent state can be derived from an observation with a projection function (for instance, a neural network as an encoder) [1], $\phi : \mathcal{O} \to \mathcal{S}$.

At time step $t$, let $o_t \in \mathcal{O}$ represent the observation composed of stacked frames, and $a_t \in \mathcal{A}$ denote the action. The dynamics can be described by the transition probability function $P$, which determines the next observation of the agent $o_{t+1} \sim P(\cdot|o_t, a_t)$ (or the next latent state of the agent

---

[1] In this paper, the concept of state representation refers to the latent state embedding that is output from the projection function, *i.e.*, $s = \phi(o)$.

$s_{t+1} \sim P(\cdot|s_t, a_t)$ in the context [2]. In this paper, we assume that the dynamics in real-world environments tend to be nearly deterministic, therefore we only focus on deterministic settings, *i.e.*, for all latent state $s \in \mathcal{S}$, $a \in \mathcal{A}$, there exists a unique $\kappa(s, a) \in \mathcal{S}$ such that $P_s^a(\kappa(s, a)) = 1$. The performance of the observation-action pair is quantified by the reward function $r(o, a) \in [R_{\min}, R_{\max}]$ provided by the environment[3]. Moreover, $\gamma$ is a discount factor ($0 < \gamma < 1$), which quantifies the value we weigh for future rewards. The agent aims to find the optimal policy $\pi(a|s)$ to maximize the expected reward $\mathbb{E}_\pi \left[ \sum_{t=0}^\infty \gamma^t r(o_t, a_t) \right]$. The learning problem becomes tractable via the projection function $\phi$ to learn a policy of the form $\pi(a|\phi(o))$.

## 2.2 BISIMULATION

In this work, we are specifically interested in preserving the inherent behavior of the states regarding task-specific information, which draws our attention to Bisimulation metrics. Bisimulation metrics were initially introduced as a pseudometric: $d : \mathcal{S} \times \mathcal{S} \to \mathbb{R}$ in Ferns et al. (2004; 2006) to measure the behavioral distance between states, which includes a reward difference term and a Wasserstein distance between transitions. Recently, Castro (2020) proposed an alternative metric known as the on-policy bisimulation ($\pi$-bisimulation) metric. Unlike the standard bisimulation metric, $\pi$-bisimulation metric focuses on behavior relative to a specific policy $\pi$:

**Theorem 1.** *($\pi$-bisimulation metric (Castro, 2020)) Define $\mathcal{F}^\pi : \mathcal{M} \to \mathcal{M}$ by*

$$\mathcal{F}^\pi(d^\pi)(s_i, s_j) = |r_{s_i}^\pi - r_{s_j}^\pi| + \gamma \mathcal{W}(d^\pi) \left( P_{s_i}^\pi, P_{s_j}^\pi \right), \tag{1}$$

*where $s_i, s_j \in \mathcal{S}$, $r_{s_i}^\pi = \sum_{a \in \mathcal{A}} \pi(a|s_i) r_{s_i}^a$, $P_{s_i}^\pi = \sum_{a \in \mathcal{A}} \pi(a|s_i) P_{s_i}^a$, and $\mathcal{W}$ is the Wasserstein distance between distributions. $\mathcal{F}^\pi$ has a least fixed point $d_\sim^\pi$, and $d_\sim^\pi$ is a $\pi$-bisimulation metric.*

The Banach fixed-point theorem can be applied to ensure the existence of a unique metric $d_\sim^\pi$, allowing us to measure the distance between distinct states via $d_\sim^\pi$. This concept has inspired subsequent research to leverage $\pi$-bisimulation metrics to shape the representations of deep RL agents (Zhang et al., 2021b; Castro et al., 2021; Zang et al., 2022). For instance, Zang et al. (2022), which learns representations by integrating cosine distance with bisimulation-based measurements, is formulated as:

$$\mathcal{F}^\pi \bar{d}(\phi^\pi(o_i), \phi^\pi(o_j)) = |r_{o_i}^\pi - r_{o_j}^\pi| + \gamma \mathbb{E}_{\substack{u \sim P_{\phi^\pi(o_i)}^\pi \\ v \sim P_{\phi^\pi(o_j)}^\pi}} [\bar{d}(u, v)], \tag{2}$$

where $\bar{d}$ represents cosine distance and $P_{\phi^\pi(o_i)}^\pi$ is the transition model on the latent embedding space. By minimizing the difference between $\bar{d}(\phi^\pi(o_i), \phi^\pi(o_j))$ and $\mathcal{F}^\pi \bar{d}(\phi^\pi(o_i), \phi^\pi(o_j))$ through a mean squared error (MSE) objective, we can obtain state representations with meaningful semantics, which can be beneficial for downstream policy training.

## 3 THEORETICAL ANALYSIS

In this section, we primarily focus on a cosine distance-based bisimulation measurement (Zang et al., 2022), highlighting potential barriers to the practical application of the bisimulation principle. Specifically, we first discuss the sufficient condition for the existence of a unique measurement $d_\sim^\pi$ based on approximate dynamics. Thereafter, we illustrate the potential issues with modeling dynamics using a Gaussian distribution. Finally, we emphasize how uninformative rewards can induce feature collapse in bisimulation objectives. The proofs of theorems are provided in Appendix B.

As aforementioned, in this paper, we mainly focus on deterministic settings, where the expectation in Equation 2 is no longer necessary. Under a system with deterministic transitions, we have the following lemma:

---

[2] For notation simplicity, we use $P_s^a$ and $r_s^a$ to denote $P(\cdot|s, a)$ and $r(s, a)$, respectively, to represent the transition and the reward function in the state space.

[3] Subsequently, we normalize the reward given by the environment to ensure that the reward utilized is definitively bounded.

**Lemma 1.** *Given a deterministic MDP, for any two states $s_i, s_j \in \mathcal{S}$, action $a \in \mathcal{A}$, and measurement $d$, we have:*

$$d(\kappa(s_i, a), \kappa(s_j, a)) = \mathcal{W}_1(d)(P^a_{s_i}, P^a_{s_j}),\tag{3}$$

*where $\kappa(s, a) \in \mathcal{S}$ is a deterministic mapping to a unique state. Besides, we further consider deterministic policies in the on-policy case, where we have:*

$$|r^\pi_{s_i} - r^\pi_{s_j}| + \gamma \mathcal{W}(d^\pi)\left(P^\pi_{s_i}, P^\pi_{s_j}\right) = |r^\pi_{s_i} - r^\pi_{s_j}| + \gamma d(\kappa(s_i, \pi), \kappa(s_j, \pi)).\tag{4}$$

As discussed in Kemertas & Aumentado-Armstrong (2021), when using an approximate forward dynamics model $\hat{P} : \mathcal{S} \times \mathcal{A} \to \mathcal{M}(\mathcal{S}')$ (where $\mathcal{M}(\mathcal{X})$ denotes the space of all probability distributions over $\mathcal{X}$), the convergence guarantees may not be applicable if compactness is not guaranteed. As a result, convergence could be problematic when the approximation error is large. We now propose a sufficient condition for a unique measurement $d^\pi_\sim$ based on approximate dynamics.

**Theorem 2** (Boundedness Condition for Convergence). *Assume $\mathcal{S}$ is compact and we have approximate dynamics $\hat{P}$, with its support being a closed subset of $\mathcal{S}$. Then, a unique bisimulation measurement $d^\pi_\sim$ of the form given in Equation 2 exists, and this measurement is bounded:*

$$\mathrm{supp}(\hat{P}) \subseteq \mathcal{S} \Rightarrow \mathrm{diam}(\mathcal{S}; d^\pi_\sim) \le \frac{1}{1-\gamma}(R_{\max} - R_{\min}),\tag{5}$$

*where $\mathrm{diam}$ is Diameter of $\mathcal{S}$.*

Following Kemertas & Aumentado-Armstrong (2021), given the approximate dynamics $\hat{P}$, we have:

**Theorem 3.** *Define $\mathcal{E}_P := \sup_{s \in \mathcal{S}} W_1(d^\pi_\sim)\left(P^\pi_s, \hat{P}^\pi_s\right)$. Then $\left\|d^\pi_\sim - \hat{d}^\pi_\sim\right\|_\infty \le \frac{2}{1-\gamma}\mathcal{E}_P$, where $\hat{d}^\pi_\sim$ is the approximate fixed point.*

Since we cannot always guarantee the condition in Theorem 2 during training, any violation of compactness in the approximate dynamics could potentially result in undesirable measurement expansion, thereby decreasing the performance. Moreover, Theorem 3 illustrates that when the error in the dynamics model is sufficiently large, it could result in a significant approximation error. These factors imply that using a Gaussian distribution to model forward dynamics may result in undesirable performance degradation when dealing with multi-modal and intricate environmental dynamics.

In addition, we discover that bisimulation-based objectives are problematic in environments with sparse rewards. Specifically, in extreme cases where the reward always remains zero, the following theorem reveals that the objective leads to a trivial solution where all state representations collapse to the same point.

**Theorem 4.** *If the reward is constantly zero, there exists a trivial solution for the bisimulation loss where all sample representations are identical, i.e., $\forall s_i, s_j \in \mathcal{S}, r^\pi_{s_i} = r^\pi_{s_j} = 0 \Rightarrow d^\pi_\sim(s_i, s_j) = 0$.*

As Theorem 4 indicates, all states are erroneously considered identical, causing the representation embedding $\phi$ to collapse accordingly. This results in the agent relinquishing all information about its underlying state. This failure case is inevitable for bisimulation-based objectives in such settings. A potential solution is to enrich the agent with additional informative knowledge, enabling it to consider not only reward-specific information but also other information pertinent to its control task.

## 4 METHOD

As aforementioned, bisimulation-based approaches have challenges regarding the limited expressive capacity of latent dynamics and inadaptability to environments with sparse rewards. To address these representational deficiencies inherent in bisimulation principles, we propose a novel representation learning method for RL, named ReBis. ReBis consists of three components: (a) mapping original observations to latent space via Siamese encoders with Block-wise masking, thereby reducing spatiotemporal redundancy; (b) constructing a transformer-based dynamics model to help agents capture multi-modal behaviors; and (c) updating representation via a reconstruction procedure in the latent space following the bisimulation principle. An overview of our method is depicted in Figure 1.

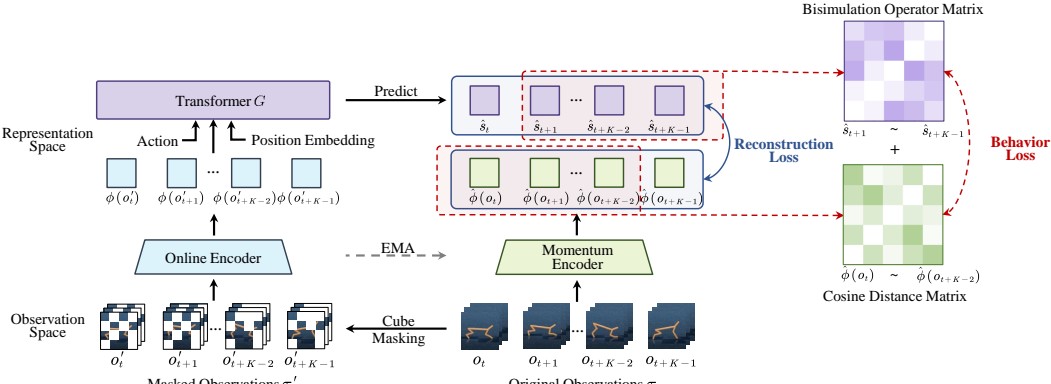

Figure 1: Overview of the ReBis framework. Masked observations and original observations are encoded through an online encoder and a momentum encoder, respectively. The transformer $G$ is then used to predict the masked content in the latent space. The reconstruction loss is measured between $K$ pairs of state representations, and the behavior loss is measured between $K-1$ state representations. Both losses are employed concurrently to train the network. The shades of color in the matrices on the right represent the range of numerical values.

**Observation Masking and Siamese Encoding.** We first consider Block-wise sampling (Wei et al., 2022), which masks visual inputs in spacetime to capture the most essential spatiotemporal information while discarding spatiotemporal redundancies. We randomly sample a consecutive sequence of K observations $\tau_K = \{o_t, o_{t+1}, \cdots, o_{t+K-1}\}$ through interactions with the environment, and stack 3 frames for each observation. We denote $\tau'_K = \{o'_t, o'_{t+1}, \cdots, o'_{t+K-1}\}$ as the masked observation sequence and $\tau_K$ as the original observation sequence. Subsequently, we utilize Siamese CNN encoder networks to project the pair of masked and original observation sequences. These weight-sharing neural networks denoted as $\phi$ and $\hat{\phi}$, are applied to two types of inputs to encode high-dimensional pixels into more task-oriented latent state representations. To prevent undesired trivial solutions, we update the parameters of the encoder network $\hat{\phi}$ with the exponential moving average (EMA) as: $\hat{\phi} \leftarrow m\hat{\phi} + (1-m)\phi$, where $m \in [0,1)$ is the momentum coefficient.

**Highly Expressive Dynamics Model.** Given that the dynamics in real-world environments tend to be nearly deterministic, expressiveness-limited dynamics, as discussed in Section 3, can lead to undesirable performance degradation. To address this, we employ a Transformer encoder as the forward model to enhance the expressiveness of the latent dynamics. Transformers have proven to be powerful (Micheli et al., 2022; Chen et al., 2022) and computationally universal (Lu et al., 2022) (even Turing Complete (Pérez et al., 2021)). They can also extensively exploit historical information (Chen et al., 2022; Micheli et al., 2023) for representation learning, aligning with the underlying settings of POMDPs. The input to the transformer encoder is the full set of tokens consisting of *state tokens*, *action tokens*, and *positional embedding*. Specifically, the masked state representation sequence $\tau_K^s = \tau_K^{\phi(o')} = \{\phi(o'_t), \phi(o'_{t+1}), \cdots, \phi(o'_{t+K-1})\}$ serves as the *state tokens*, while the corresponding embedded action sequence $\tau_K^a = \tau_K^{\psi(a)} = \{\psi(a_t), \psi(a_{t+1}), \cdots, \psi(a_{t+K-1})\}$ is used as the *action tokens* where $\psi$ is the embedding layer that projects actions to the same feature dimension as $\phi(o')$. We also add standard relative position embeddings to both token sequences (state and action tokens), which is denoted as $\tau_K^p$. After feeding all tokens into a Transformer encoder $G$, the output tokens, defined as $\hat{\tau}_K^s = \{\hat{s}_t, \hat{s}_{t+1}, \cdots, \hat{s}_{t+K-1}\}$, where $\hat{s}_{t+1} := \kappa(\phi(o_t), a_t)$, are the predictive reconstruction results for the latent representations (see Appendix F.2 for more details). Hence, the ability of the transformer architecture to model long-range dependencies and learn inherent uncertainties within the environment serves a dual purpose. It not only retains control-centric reward-free information by leveraging a masking scheme, but also functions as an implicit dynamic model. This dual functionality promotes sample efficiency and enhances overall model performance, proving valuable in tackling image-based RL or POMDPs.

**Learning Objective.** We use the encoded representations from the original unmasked observation sequence as the targets for reconstruction and prediction. Employing the transformer encoder as a highly expressive dynamics model, we first define a bisimulation-based update operator as below.

**Definition 1.** *Given policy $\pi$, we define the update operator as*

$$\mathcal{F}^\pi \bar{d}(\hat{\phi}(o_i), \hat{\phi}(o_j)) = |r^\pi_{o_i} - r^\pi_{o_j}| + \gamma \bar{d}(\kappa(\phi(o_i), \pi), \kappa(\phi(o_j), \pi)), \quad (6)$$

*where $\kappa$ is exact the transformer G that we used, and rewards can be sampled from the underlying signals provided by the environment.*

Accordingly, we can minimize the following behavioral loss to capture the behavioral characteristics that contain reward information of different state representations, given as:

$$\mathcal{L}_{\text{behavior}} = \text{MSE}\left(\bar{d}(\hat{\phi}(o_i), \hat{\phi}(o_j)), \mathcal{F}^\pi \bar{d}(\hat{\phi}(o_i), \hat{\phi}(o_j))\right). \quad (7)$$

To integrate temporal information from observation sequences and enhance the expressive power of the state representations, the latent reconstruction loss is formulated as the mean squared error loss between original state representations and their predicted reconstructions in the latent space:

$$\mathcal{L}_{\text{reconstruction}} = \text{MSE}(\tau_K^{\hat{\phi}(o)}, \hat{\tau}_K^s), \quad (8)$$

where $\hat{\tau}_K := (\hat{\tau}_K^s, \hat{\tau}_K^a) = G(\tau_K^s, \tau_K^a, \tau_K^p)$.

To concurrently optimize both the behavioral loss and the latent reconstruction loss, the overall loss function of ReBis is formulated as:

$$\mathcal{L} = \mathcal{L}_{\text{behavior}} + \beta \mathcal{L}_{\text{reconstruction}}, \quad (9)$$

where $\beta$ weighs the importance between $\mathcal{L}_{\text{behavior}}$ and $\mathcal{L}_{\text{reconstruction}}$. Note that, our objective does not make any assumptions about Gaussianity and can benefit from the strong expressive capabilities of the transformer architecture. In addition, we also find that the dynamics model can function as an asymmetric module in the Siamese architecture to prevent potential feature collapse in environments with uninformative rewards. The following theorem proves how such an asymmetrical architecture alleviates feature collapse by increasing the effective feature dimensionality throughout the training.

**Theorem 5.** *Under mild data assumptions as in Zhuo et al. (2023), each gradient update of the reconstruction loss $\mathcal{L}_{reconstruction}$ improves the effective dimensionality of output features $\hat{\tau}_K^s$.*

**Summary.** Our proposed self-supervised auxiliary objective enables the learned state representations to effectively capture how an agent interacts with the environment. By perceiving useful spatiotemporal information and distinguishing the behavior differences between states, the agent is able to learn control-centric representations that facilitate policy learning. Serving as a plug-and-play representation learning module, ReBis can be readily integrated into any off-the-shelf downstream RL objectives to improve the agent's understanding of the environment.

## 5 EXPERIMENTS

This section evaluates the sample efficiency and asymptotic performance of our proposed method on two commonly used benchmarks, including Atari 2600 Games (Bellemare et al., 2013) for discrete control and DeepMind Control Suite (DMControl) (Tassa et al., 2018) for continuous control. To further assess the capability of our model ReBis on capturing task-specific information, we evaluated its performance in more complex and realistic scenarios, where we introduced disturbances by replacing the background with natural videos (Zhang et al., 2018). The ablation study and all experimental results are included in the Appendix D.3

### 5.1 IMPLEMENTATION DETAILS

**Atari 2600 Games.** As a representation learning approach, ReBis can be integrated into any type of downstream RL algorithm. For the experiments, we chose Rainbow (Hessel et al., 2018) as the

downstream RL agent. We trained and evaluated the model on the Atari-100k benchmark, which comprises 26 Atari games and allows 100k interaction steps (or 400K frames with a frame skip of 4) for training. The Human-normalized Score (HNS) was employed to measure the performance in each game. We followed the setting in Agarwal et al. (2021a) to evaluate overall performance with robust and efficient aggregate metrics, including the interquartile mean (IQM) and optimality gap (OG), with 95% confidence intervals (CIs), for a more rigorous assessment on high-variance benchmarks with limited runs. All experiment results on Atari games are based on 3 random seeds.

**DMControl with the default setting.** The DMControl is a suite of continuous control tasks, which are powered by the MuJoCo physics engine (Todorov et al., 2012) and rendered using raw pixels. We chose Soft Actor-Critic (Haarnoja et al., 2018) as the downstream RL agent and experimented on 11 environments from DMControl to evaluate the performance of ReBis, including complex dynamics, sparse rewards, and hard exploration. We reported mean and std numerical results across 10 episodes at 500k environment steps, which are denoted as DMControl-500k benchmarks. The score for each environment ranges from 0 to 1000. All experimental results on DMControl tasks are based on 5 random seeds.

**DMControl task with Distractions.** To assess the robustness of ReBis on tasks with more realistic observations, we modified existing reinforcement learning tasks in DMControl to incorporate natural signals.

In the experiments, we replaced the default simple backgrounds with natural videos from the Kinetics dataset (Kay et al., 2017), inserting them as the background of observations in DMControl tasks (see Figure 2 for examples). Specifically, agents were trained in default environments without any background distractions and were expected to generalize to novel environments with natural video distractions. These settings significantly expand the observation space of the environments, presenting a complex challenge in effectively concentrating on task-related objects while ignoring visually distracting elements within the scenes. In this experiment, we compared the averaged scores across ten episodes at 500k environment steps over three random seeds.

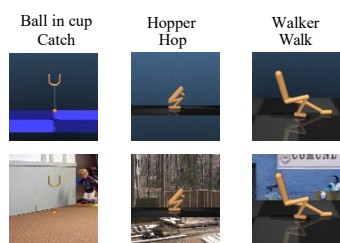

Figure 2: Pixel observations in DMControl tasks in the default settings (top row) and in the natural video distraction settings (bottom row).

## 5.2 EXPERIMENT RESULTS

**Results on Atari-100k.** We evaluated the performance of ReBis in comparison with various methods, including MLR (Yu et al., 2022), SimSR (Zang et al., 2022), PlayVirtual (Yu et al., 2021), SPR (Schwarzer et al., 2020), DrQ (Yarats et al., 2021b), DrQ($\epsilon$) (DrQ using the $\epsilon$-greedy parameters in (Castro et al., 2018)), CURL (Laskin et al., 2020), OTR(Kielak, 2020), and DER (Van Hasselt et al., 2019), all are incorporated with Rainbow (Hessel et al., 2018)

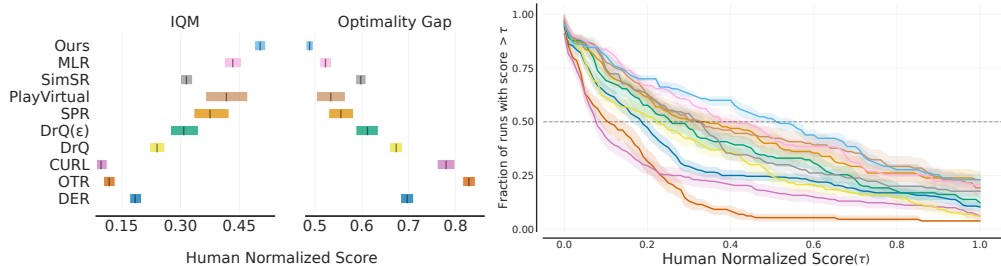

Figure 3: (**Left**) Results on Atari-100k over 3 seeds. Aggregate metrics (IQM and OG) with 95% confidence intervals were used for the evaluation. Higher IQM and lower OG are better. (**Right**) Performance profiles on the Atari-100k benchmark based on human-normalized score distributions. Shaded regions indicate 95% confidence bands.

| DMControl-500k | CURL | DrQ | PlayVirtual | MLR | SimSR | Ours |
|---|---|---|---|---|---|---|
| Ball in cup, Catch | 950±38 | 965±17 | 976±16 | 975±6 | 951±26 | **982±9** |
| Cartpole, Swingup | 822±67 | 864±35 | 874±17 | 875±11 | 846±49 | **883±26** |
| Cartpole, Swingup Sparse | 0±0 | 0±0 | 112±9 | 67±27 | 103±59 | **518±45** |
| Cheetah, Run | 555±110 | 663±54 | 729±30 | 697±56 | 725±59 | **748±44** |
| Finger, Spin | 920±41 | 934±131 | 965±40 | 969±28 | 964±20 | **971±26** |
| Finger, Turn Easy | 293±17 | 365±21 | 339±25 | 374±32 | 435±14 | **652±34** |
| Finger, Turn Hard | 91±19 | 138±31 | 194±42 | 201±28 | 239±16 | **328±35** |
| Hopper, Hop | 12±8 | 116±78 | 133±29 | 134±8 | 200±29 | **233±13** |
| Hopper, Stand | 640±110 | 809±66 | 896±36 | 901±34 | 858±68 | **927±18** |
| Pendulum, Swingup | 242±36 | 345±25 | 381±38 | 434±27 | 446±8 | **458±7** |
| Walker, Walk | 909±48 | 910±73 | 934±49 | 928±33 | 935±4 | **941±21** |

Table 1: Results (mean ± std) on the DMControl-500k benchmarks with **default** settings. The environments marked in blue color are sparse reward environments.

for policy training. In Figure 3, ReBis attains the highest IQM score of **0.501** and the lowest OG of **0.488**, showing the effectiveness of ReBis in prompting the downstream policy performance. Notably, our approach achieves the highest scores in **16/26** games, indicating that our approach can indeed improve the perception of the agent by better capturing control-centric information. The full scores of ReBis across the 26 Atari games and more comparisons and analysis can be found in Appendix D.1.

**Results on DMControl with default settings.** Under default settings, we evaluated the performance of ReBis against sample-efficient model-free RL methods with an additional focus on effective representation learning of states/observations, such as CURL, DrQ, PlayVirtual, MLR, and SimSR. As shown in Table 1, ReBis surpasses previous methods on DMControl-500k across all representative tasks. For challenging tasks with sparse rewards, such as *Ball in cup Catch*, *Cartpole Swingup Sparse*, *Finger turn easy*, *Finger turn hard*, and *Pendulum, Swingup*, the effectiveness of ReBis in complex environments further underscores our method's ability to capture agent dynamics by focusing on reward and temporal information. Regarding sample efficiency, the results of DMControl-100k and the learning curves are provided in Appendix D.2.

## 5.3 CAN REBIS CAPTURE CONTROL-CENTRIC INFORMATION?

In our pursuit to extract control-centric insights from visually noisy real-world signals, we evaluated the performance of ReBis in the environments with background distractions. Real-world visual data often contains redundancy and control-irrelevant elements, which motivated our investigation. Table 2 summarizes our findings, revealing that MLR's performance deteriorates in the presence of strong distractions, while SimSR fares better but still experiences a decline. In contrast, ReBis maintains remarkable stability across tasks, particularly excelling in sparse reward environments such as *Ball in cup, Catch*, *Cartpole Swingup Sparse*, *Finger turn easy/hard*, and *Pendulum, Swingup*. The results suggest that ReBis effectively filters out task-irrelevant information with complex environments.

To ascertain the extent of our model's capability in filtering background redundancy and focusing on control-centric features, we employed the Grad-CAM (Selvaraju et al., 2017) for feature visualization. This approach allowed us to delve into the inner workings of ReBis and gain insights into its effectiveness in capturing task-relevant information and extracting pertinent features. Our analysis was conducted on three sparse environments of varying difficulty levels of background distractions.

The heatmaps shown in Figure 4, generated using Grad-CAM, demonstrate that ReBis is able to reduce background noise and identify features relevant to control. This observation validates that ReBis can effectively extract control-centric information from visual inputs containing noise.

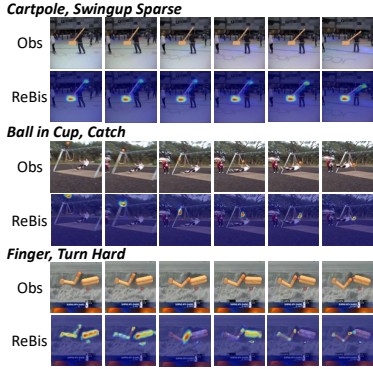

Figure 4: The feature visualization of our learned representations using Grad-CAM.

| DMControl-unseen | CURL | DrQ | PlayVirtual | MLR | SimSR | Ours |
|---|---|---|---|---|---|---|
| Ball in cup, Catch | 316±92 | 318±75 | 815±102 | 832±76 | 894±35 | **970±16** |
| Cartpole, Swingup | 335±17 | 363±39 | 662±152 | 845±36 | 697±73 | **859±21** |
| Cartpole, Swingup Sparse | 0±0 | 0±0 | 22±2 | 21±2 | 25±3 | **216±15** |
| Cheetah, Run | 162±16 | 266±35 | 539±25 | 401±14 | 602±35 | **712±30** |
| Finger, Spin | 396±29 | 404±17 | 763±52 | 882±13 | 563±69 | **893±35** |
| Finger, Turn Easy | 2±1 | 15±3 | 104±22 | 292±51 | 376±18 | **559±42** |
| Finger, Turn Hard | 0±0 | 0±0 | 106±15 | 184±32 | 182±29 | **236±25** |
| Hopper, Hop | 9±3 | 14±4 | 27±7 | 20±6 | 135±25 | **145±28** |
| Hopper, Stand | 319±176 | 423±95 | 473±63 | 794±62 | 505±112 | **881±26** |
| Pendulum, Swingup | 27±8 | 41±13 | 123±19 | 190±11 | 204±39 | **255±14** |
| Walker, Walk | 502±75 | 616±48 | 595±17 | 884±25 | 673±18 | **893±41** |

Table 2: Results (mean ± std) on the DMControl-500k with **unseen** background distractions, i.e., training the agent on the default setting and evaluating it on tasks with natural video distractions. The environments marked in blue color are sparse reward environments.

| Algorithms | DrQ | CURL | PlayVirtual | MLR | SimSR | Ours |
|---|---|---|---|---|---|---|
| Exogenous Invariant | ✗ | ✗ | ✗ | ✗ | ✓ | ✓ |
| Reward Aware | ✗ | ✗ | ✗ | ✗ | ✓ | ✓ |
| Dynamics Recovery | ✗ | ✗ | ✓ | ✓ | ✓ | ✓ |
| Feasibility to sparse reward tasks | ✓ | ✓ | ✓ | ✓ | ✗ | ✓ |

Table 3: **Overview of Properties** of prior approaches on model-free representation learning in RL. The comparison to ReBis aims to be as generous as possible to the baselines. ✗ indicates a known counterexample for a given property. We compare four different properties.

## 6 RELATED WORK

In RL, the goal of effective state representation learning is to learn a mapping function that translates rich, high-dimensional observations into a compact latent space. Recent research has explored representation learning in RL from various perspectives. A prevalent approach, CURL (Laskin et al., 2020), learns a representation that is invariant to a class of data augmentations. However, it fails to capture either control-centric information or reward-relevant knowledge. Similarly, DrQ (Yarats et al., 2021b), which heavily relies on data augmentation strategies, struggles to account for exogenous noise. Self-supervised objectives, based on visual input and sequential interaction, have been introduced by PlayVirtual (Yu et al., 2021). Recently, mask-based methods (Seo et al., 2022; Yu et al., 2022), which have been proposed to reduce spatiotemporal redundancy in particular, recover latent dynamics by constructing a transformer model. However, these methods consistently overlook the importance of reward signals. In contrast, bisimulation-based methods, such as Castro et al. (2021); Zang et al. (2022), are fully reward-aware, but may disregard critical spatiotemporal information. Although this information is not directly related to rewards, it is essential for control determination in environments with uninformative rewards. In contrast to these methods, ReBis addresses these shortcomings by learning control-centric representations while maintaining reward awareness, and effectively eliminating spatiotemporal redundancy. Table 3 presents a comprehensive overview of these representative prior approaches from four perspectives.

## 7 DISCUSSION

In this paper, we analyze the bound and the potential harm of the previous objectives that follows bisimulation principles, emphasizing the necessity for a highly expressive dynamics model and spatiotemporal knowledge in sparse reward environments. Therefore, we present ReBis as an effective way of learning state representations tailored to vision-based RL. The empirical results demonstrate the superiority of the representations produced by ReBis. One potential limitation of our approach is its time complexity during deployment, as it includes transformer architecture in the module, similar to the previous state-of-the-art methods such as MLR (Yu et al., 2022) (Time complexity comparison can be found in Appendix F.4). An alternative way to address this issue is by applying our approach to offline settings, allowing for the pretraining of the encoder.

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

## A ADDITIONAL RELATED WORK

The success of deep RL in many fields is due in part to the utilization of the state representation learning. Traditionally, state representation learning has been framed as learning state abstractions / aggregations (Andre & Russell, 2002; Ferns et al., 2006; Mannor et al., 2004; Comanici et al., 2012), and most of these methods aim to reduce the original state space size and to minimize the system complexity. Recent studies present promising results in learning robust representations that can then be used to accelerate policy learning in pixel-based observation spaces. Lee et al. (2020); Yarats et al. (2021c) propose to train deep auto-encoders with a reconstruction loss to learn compact representations, Shelhamer et al. (2017); Hafner et al. (2019b) learn state representations from predictive losses to maintain trajectory consistency, van den Oord et al. (2018); Laskin et al. (2020); Stooke et al. (2021) use contrastive losses as auxiliary tasks to improve performance, Xu et al. (2014); Krishnamurthy et al. (2016); Yarats et al. (2021a) develop state clustering methods to improve sample-efficiency, and finally Zhang et al. (2021a); Castro et al. (2021); Zang et al. (2022) measure state distance by bisimulation similarity to shape state representations.

## B PROOF

**Lemma 1.** *Given a deterministic MDP, for any two states $s_i, s_j \in \mathcal{S}$, action $a \in \mathcal{A}$, and measurement $d$, we have:*

$$d(\kappa(s_i, a), \kappa(s_j, a)) = \mathcal{W}_1(d)(P_{s_i}^a, P_{s_j}^a) \tag{10}$$

*Proof.* Given two state distributions $X, Y \in \Delta(\mathcal{S})$ [4] and a measurement $d \in \mathcal{M}$, the Wasserstein $\mathcal{W}_1(d)(X, Y)$ can be expressed by the following (primal) linear program (LP), which "lifts" a measurement $d \in \mathcal{M}$ onto one in the set of all metrics between probability distributions over $\mathcal{S}$ Villani (2016):

$$\max_{\mathbf{u}} \in \mathbb{R}^{|\mathcal{S}|} \sum_{s \in \mathcal{S}} (X(s) - Y(s)) \, u_s$$
$$\forall s, s' \in \mathcal{S}, \ u_s - u_{s'} \leq d(s, s')$$
$$0 \leq \mathbf{u} \leq 1$$

It can be expressed in its dual form:

$$\min_{\boldsymbol{\lambda}} \sum_{s_i', s_j' \in \mathcal{S}} \lambda_{s_i', s_j'} d(s_i', s_j')$$
$$\text{s.t.} \quad \forall s_i' \in \mathcal{S}, \quad \sum_{s_j'} \lambda_{s_i', s_j'} = P_{s_i}^a(s_i')$$
$$\forall s_i' \in \mathcal{S}, \quad \sum_{s_i'} \lambda_{s_i', s_j'} = P_{s_j}^a(s_j') \tag{11}$$
$$\boldsymbol{\lambda} \geq 0$$

By the deterministic assumption it follows that $\lambda_{s_i', s_j'} = 0$ whenever $s_i' \neq \kappa(s_i, a)$ or $s_j' \neq \kappa(s_j, a)$, which means that only $\lambda_{\kappa(s_i,a),\kappa(s_j,a)}$ is positive. By the equality constraints it then follows that $\lambda_{\kappa(s_i,a),\kappa(s_j,a)} = 1$, resulting in $d(\kappa(s_i, a), \kappa(s_j, a))$ as the minimal objective value. □

**Theorem 2** (Boundedness Condition for Convergence). *Assume $\mathcal{S}$ is compact and we have approximate dynamics $\hat{P}$, with its support being a closed subset of $\mathcal{S}$. Then, a unique bisimulation measurement $d_\sim$ of the form given in Equation 6 exists, and this measurement is bounded:*

$$\text{supp}(\hat{P}) \subseteq \mathcal{S} \Rightarrow \text{diam}(\mathcal{S}; d_\sim) \leq \frac{1}{1 - \gamma}(R_{\max} - R_{\min}). \tag{12}$$

*where* diam *is Diameter of $\mathcal{S}$.*

---

[4]$\Delta(\mathcal{S})$ is the Laplace–Beltrami operator on submanifolds in state space $\mathcal{S}$.

*Proof.* This mimics the proof of Lemma 1 in Zang et al. (2022), except it replaces $P$ with an approximate dynamics model $\hat{P}$. First, let $\phi, \phi' : \mathcal{O} \to \mathcal{S}$, denote $\hat{d}(\phi(o_i), \phi(o_j))$ as $d(s_i, s_j)$, and $\hat{d}(\phi'(o_i), \phi'(o_j))$ as $d'(s_i, s_j)$, then we have:

$$\mathcal{F}^\pi d(s_i, s_j) - \mathcal{F}^\pi d'(s_i, s_j) \leq \gamma ||d - d'||_\infty, \tag{13}$$

when considering the transition as a stochastic matrix. This implies $\mathcal{F}^\pi$ is a $\gamma$-contraction. Then, as noted in Chen et al. (2012); Ferns & Precup (2014), the set of couplings for two probability functions forms a polytope while the Wessertain distance optimizes a linear function over vertices of the polytope. Follow Lemma 5 in Kemertas & Aumentado-Armstrong (2021), we have:

$$\text{supp}(\hat{P}) \subseteq \mathcal{S} \Rightarrow \sup_{\mathbf{s}_i, \mathbf{s}_j \in \mathcal{S} \times \mathcal{S}} d_\sim \left( \hat{P}^\pi_{s_i}, \hat{P}^\pi_{s_j} \right) \leq \text{diam}\left( \mathcal{S}; d_\sim \right), \forall p \geq 1. \tag{14}$$

We further assume that the reward function is an unbiased estimator over the latent states, *i.e.*, $r^\pi_{o_i} = r^\pi_{s_i}$, where $s_i = \phi(o_i)$. Then, by integrating Equation 6, for the fixed point, we have

$$\begin{aligned} d_\sim(s_i, s_j) &= |r^\pi_{s_i} - r^\pi_{s_j}| + \gamma d_\sim \left( \hat{P}^\pi_{s_i}, \hat{P}^\pi_{s_j} \right), \\ &\leq (R_{\max} - R_{\min}) + \gamma \, \text{diam}\left( \mathcal{S}; d_\sim \right), \end{aligned} \tag{15}$$

which implies,

$$\begin{aligned} \text{diam}\left( \mathcal{S}; d_\sim \right) &\leq (R_{\max} - R_{\min}) + \gamma \, \text{diam}\left( \mathcal{S}; d_\sim \right) \\ &\leq \frac{1}{1 - \gamma}(R_{\max} - R_{\min}). \end{aligned} \tag{16}$$

$\square$

**Theorem 3.** *Define* $\mathcal{E}_P := \sup_{s \in \mathcal{S}} W_1(d_\sim)\left( P^\pi_s, \hat{P}^\pi_s \right)$. *Then* $\left\| d_\sim - \tilde{d}_\sim \right\|_\infty \leq \frac{2}{1-\gamma}\mathcal{E}_P$, *where* $\tilde{d}_\sim$ *is the approximate fixed point.*

*Proof.* First, we have the following lemma:

**Lemma 4.** *(Kemertas & Aumentado-Armstrong, 2021) Assume* $\text{supp}(\hat{P}) \subseteq \mathcal{S}$ *and* $1 - \gamma a_p > 0$. *Then*

$$||d_\sim - \tilde{d}_\sim||_\infty \leq \frac{2}{1 - \gamma a_p}\mathcal{E}_r + \frac{2\gamma}{1 - \gamma a_p}\mathcal{E}_P + \frac{\gamma[a_p - 1]}{1 - \gamma a_p}\text{diam}(\mathcal{S}; d_\sim) \tag{17}$$

*where* $a_p = 2^{(p-1)/p}$ *and* $\text{diam}(\mathcal{S}; d_\sim) \leq \frac{1}{1-\gamma}(R_{\max} - R_{\min})$.

Then let $p = 1$, we have $a_p = a_1 = 1$, and we have:

$$\left\| d_\sim - \tilde{d}_\sim \right\|_\infty \leq \frac{2}{1 - \gamma}\mathcal{E}_P. \tag{18}$$

$\square$

**Theorem 4.** *If the reward is constantly zero, there exists a trivial solution for the bisimulation loss where all sample representations are identical, i.e.,* $\forall s_i, s_j \in \mathcal{S}, r^\pi_{s_i} = r^\pi_{s_j} = 0 \Rightarrow d^\pi_\sim(s_i, s_j) = 0$.

*Proof.* For the fixed point $d^\pi_\sim$, we have:

$$\begin{aligned} d^\pi_\sim(s_i, s_j) &= \mathcal{F}^\pi d^\pi_\sim(s_i, s_j) = |r^\pi_{s_i} - r^\pi_{s_j}| + \gamma d^\pi_\sim(s'_i, s'_j) \\ &\leq |r^\pi_{s_i} - r^\pi_{s_j}| + \gamma \max_{s'_i, s'_j} d^\pi_\sim(s'_i, s'_j) \\ &\leq |r^\pi_{s_i} - r^\pi_{s_j}| + \gamma \max_{s'_i, s'_j}(|r^\pi_{s'_i} - r^\pi_{s'_j}| + \gamma \max_{s''_i, s''_j} d^\pi_\sim(s''_i, s''_j)) \\ &\leq \max_{x,y}|R^\pi_x - R^\pi_y| + \gamma \max_{s'_i, s'_j}(\max_{x,y}|R^\pi_x - R^\pi_y| + \gamma \max_{s''_i, s''_j} d^\pi_\sim(s''_i, s''_j)) \\ &\leq \dots \end{aligned} \tag{19}$$

As we can apply the inequality recursively and substitute the inequality back into the original inequality, we can have an infinite series that becomes a sum of the largest rewards differences scaled by powers of $\gamma$. Accordingly, we have:

$$d_\sim^\pi(s_i, s_j) \leq \frac{\max_{x,y} |R_x^\pi - R_y^\pi|}{1 - \gamma}. \tag{20}$$

When the reward is constantly zero, the reward difference $\max_{x,y} |R_x^\pi - R_y^\pi|$ equals to zero everywhere, and thus we have $d_\sim^\pi(s_i, s_j) \leq 0$, as the fact that $d_\sim^\pi(s_i, s_j) \geq 0$ by definition, we finally have $d_\sim^\pi(s_i, s_j) = 0$. $\qquad\qquad\square$

Here, we give a formal version of Theorem 5, and then provide its proof. Here, to distinguish them, we call the encoder with gradient as online encoder and the momentum encoder without gradient as the target encoder.

**Theorem 5.** *For the simplicity of theoretical exposure, we follow the framework in Tian et al. (2021); Zhuo et al. (2023) and adopt the following simplified setting for our analysis:*

- *the encoder network is a linear layer $\hat{\tau}_x = W_f x, W_f \in \mathbb{R}^{d \times k}$, and the two branches share the same encoder module (ignoring the momentum);*

- *we consider Gaussian noise as the data augmentation operation, where the original data $\bar{x} \sim \mathcal{N}(0, D)$ follows a zero-mean Gaussian distribution with a diagonal covariance matrix $D \in \mathbb{R}^n$ whose diagonal elements $d_1 \geq \cdots \geq d_n$. Accordingly, the data is generated from $\mathcal{A}(x|\bar{x}) \sim N(\bar{x}, \sigma^2 I)$. The augmented data serve as inputs of both branches;*

- *the Transformer module acts as a linear layer, a $\tau_x = W\hat{\tau}_x, W \in \mathbb{R}^{k \times k}$.*

*For a specific feature representation, e.g., $\tau_x$, we measure its effective dimensionality by the effective rank of its correlation matrix. Specifically, the correlation matrix is $C_{\tau_x} = \mathbb{E}_x \tau_x \tau_x^\top$ with eigendecomposition $C_{\tau_x} = V\Lambda V^\top$, where $V$ is the eigenvector and $\Lambda = \mathrm{diag}(\lambda_1, \ldots \lambda_k)$ is the diagonal eigenvalue matrix, and effective rank is defined as the Shannon entropy of the normalized eigenvalues, i.e., $\mathrm{erank}(C_{\tau_x}) = \mathcal{H}(p) = -\sum_{i=1}^k p_i \log p_i$, where $p_1 = \lambda_1 / \sum_i \lambda_i$.*

*Then, after one gradient descent step of the MSE reconstruction loss (Equation 8) from the $t$-th step to the $(t+1)$-th step, the effective feature dimensionality is improved upon during this process, and thus prevent potential feature collapse. Formally,*

$$\mathrm{erank}(C_{\tau_x}^{(t+1)}) > \mathrm{erank}(C_{\tau_x}^{(t)}). \tag{21}$$

*Proof.* Here, we outline the main steps of the proof. First, following Lemma 1 from Zhuo et al. (2023), it is easy to show that the online output features $\tau_x$ and the target output features $\hat{\tau}_x$ share the same eigenspace $V$ during the reconstruction loss. This fact means that *the proposed Transformer module essentially behaves as a low-pass filter* that is learned to reconstruct the original inputs from the masked noisy ones. Formally, we can derive that the closed-form solution of the predictor in the form of a spectral filter, i.e., $W = VSV^\top$, where $V$ is the eigenvector of $C_{\tau_x}$, and $S = \mathrm{diag}(s_1, \ldots, s_k)$,

$$s_i = \sqrt{\frac{d_i}{d_i + \sigma}}, \quad i = 1, \ldots, k, \tag{22}$$

is monotonically decreasing w.r.t. $i$ because $d_1 \geq \cdots \geq d_k$, meaning that $W$ is a low-pass spectral filter. Therefore, applying Theorem 4 in Zhuo et al. (2023), we can conclude that the effective rank will improve after the gradient descent step of the MSE loss. $\qquad\square$

## C COMPARISON

### C.1 WORLD MODEL

The most commonly referenced world models are those employing RSSM (Recurrent State-Space Model), including Dreamer(Hafner et al., 2019a), DreamerV2(Hafner et al., 2020), DreamerV3(Hafner et al., 2023), TIA(Fu et al., 2021) and others. These approaches predominantly fall

under the category of model-based techniques, which learn a latent dynamics model with both deterministic and stochastic components. Notably, learning to reconstruct the reward and the observation explicitly is a key component of these models. While these approaches and ours are different in two aspects: i)ReBis do not reconstruct the observation, instead, we mask the observation inputs and reconstruct the latent feature of the states; ii)ReBis is totally model-free, as the reward model and explicit transition model are unnecessary.

## C.2 SIMSR

Among the series of bisimulation-based approaches, SimSR has achieved commendable results. However, it has certain limitations. First, it may encounter challenges in sparse reward settings as the objective entirely depends on the reward function/model. Furthermore, from the model architecture perspective, it introduces additional bias by explicitly modeling the latent dynamic model. In contrast, to enhance the perception ability, we turn to the utilization of the masking strategy and the transformer structure to incorporate spatiotemporal information, which we have proven is theoretically grounded in Theorem 5.

## C.3 MLR

MLR is an approach that employs a masking strategy to reconstruct original image features, typically regarded as an auxiliary task. It prioritizes the extraction of inherent information from images and often disregards task-related information. From a model architecture perspective, MLR utilizes a complex network structure, including two convolutional neural networks (CNN), a Transformer, two projection heads, and one prediction head. In contrast to MLR, our modelReBis is fundamentally rooted in the principles of bisimulation, therefore tailoring its objectives and training process to benefit the bisimulation. This design allows our model to strike a favorable balance between efficiency and capability, especially in capturing control-centric (reward-associated) state dynamics. In comparison, MLR incorporates additional projection and prediction heads, contributing to increased complexity and computational demands. Furthermore, MLR neglects reward-related information, limiting its ability to capture state dynamics effectively.

# D ALL EXPERIMENTAL RESULTS

## D.1 PERFORMANCE ON ATARI

As shown in Table 4, we compared the results of ReBis across all 26 games with the SOTA methods on the Atari-100k benchmark based on 3 random seeds. In the evaluation, the ReBis reaches the highest scores on 16 out of 26 games and outperforms the SOTA methods on the interquartile-mean (IQM) and optimality gap (OG) with 95% confidence intervals (CIs). To further demonstrate the effectiveness of ReBis, we also compared the human-normalized scores (HNS) with 95% CIs in Figure 5, demonstrating the effectiveness of our method. The results of DER (Van Hasselt et al., 2019), OTR (Kielak, 2020), CURL (Laskin et al., 2020), DrQ(Yarats et al., 2021b) and SPR (Schwarzer et al., 2020), are from rliable (Agarwal et al., 2021b), based on 100 random seeds. The results of PlayVirtual(Yu et al., 2021) are based on 15 random seeds. The results of MLR (Yu et al., 2022) and SimSR(Zang et al., 2022), are based on 3 random seeds.

## D.2 PERFORMANCE ON DMCONTROL

Beyond the main manuscript's numerical results in Table 1, we also conducted an analysis over 100k interaction steps in the DMControl task in Table 5. In terms of sample efficiency, ReBis demonstrates notable superiority, as evidenced by its performance at the 100k interaction steps mark. Particularly for challenging tasks characterized by sparse rewards, our method achieves positive rewards within a reduced number of interaction steps.

We draw the convergence curves between ReBis and other methods in Figure 6 based on 5 random seeds for each environment. The curves reveal that ReBis can achieve convergence faster than other methods in all 9 environments and finally obtain the highest reward, demonstrating its effectiveness and efficiency. Furthermore, ReBis converged much faster than SimSR and other methods in

| Game | Human | Random | DER | OTR | CURL | DrQ | SPR | PlayVirtual | MLR | SimSR | Ours |
|---|---|---|---|---|---|---|---|---|---|---|---|
| Alien | 7127.7 | 227.8 | 802.3 | 570.8 | 711 | 734.1 | 841.9 | 947.8 | 990.1 | 756.1 | **1028.9** |
| Amidar | 1719.5 | 5.8 | 125.9 | 77.7 | 113.7 | 94.2 | 179.7 | 165.3 | 227.7 | 135.9 | **239.4** |
| Assault | 742 | 222.4 | 561.5 | 330.9 | 500.9 | 479.5 | 565.6 | 702.3 | 643.7 | 622.5 | **726.3** |
| Asterix | 8503.3 | 210 | 535.4 | 334.7 | 567.2 | 535.6 | **962.5** | 933.3 | 883.7 | 853.3 | 943.3 |
| Bank Heist | 753.1 | 14.2 | 185.5 | 55 | 65.3 | 153.4 | **345.4** | 245.9 | 180.3 | 100 | 294.7 |
| Battle Zone | 37187.5 | 2360 | 8977 | 5139.4 | 8997.8 | 10563.6 | 14834.1 | 13260 | 16080 | 12571 | **16797** |
| Boxing | 12.1 | 0.1 | -0.3 | 1.6 | 0.9 | 6.6 | 35.7 | **38.3** | 26.4 | 2 | 38 |
| Breakout | 30.5 | 1.7 | 9.2 | 8.1 | 2.6 | 15.4 | 19.6 | **20.6** | 16.8 | 12.5 | 20 |
| Choppe Cmd | 7387.8 | 811 | 925.9 | 813.3 | 783.5 | 792.4 | 946.3 | 922.4 | 910.7 | 820 | **983** |
| Crazy Climber | 35829.4 | 10780.5 | 34508.6 | 14999.3 | 9154.4 | 21991.6 | **36700.5** | 23176.7 | 24633.3 | 23695 | 25385 |
| Demo Attack | 1971 | 152.1 | 627.6 | 681.6 | 646.5 | 1142.4 | 517.6 | **1131.7** | 854.6 | 1093 | 1063.5 |
| Freeway | 29.6 | 0 | 20.9 | 11.5 | 28.3 | 17.8 | 19.3 | 16.1 | 30.2 | 29.7 | **31.5** |
| Frostbite | 4334.7 | 65.2 | 871 | 224.9 | 1226.5 | 508.1 | 1170.7 | 1984.7 | 2381.1 | 1653 | **2614.8** |
| Gopher | 2412.5 | 257.6 | 467 | 539.4 | 400.9 | 618 | 660.6 | 684.3 | 822.3 | 792 | **824.9** |
| Hero | 30826.4 | 1027 | 6226 | 5956.5 | 4987.7 | 3722.6 | 5858.6 | **8597.5** | 7919.3 | 6552 | 8575 |
| Jamesbond | 302.8 | 29 | 275.7 | 88 | 331 | 251.8 | 366.5 | 394.7 | 423.2 | 401 | **437.7** |
| Kangaroo | 3035 | 52 | 581.7 | 348.5 | 740.2 | 974.5 | 3617.4 | 2384.7 | 8516 | 8426 | **8632.3** |
| Krull | 2665.5 | 1598 | 3256.9 | 3655.9 | 3049.2 | 4131.4 | 3681.6 | 3880.7 | 3923.1 | 3639 | **4444.7** |
| Kung Fu Master | 22736.3 | 258.5 | 6580.1 | 6659.6 | 8155.6 | 7154.5 | **14783.2** | 14259 | 10652 | 9986 | 11406 |
| Ms Pacman | 6951.6 | 307.3 | 1187.4 | 908 | 1064 | 1002.9 | 1318.4 | 1335.4 | 1481.3 | 1128 | **1496.1** |
| Pong | 14.6 | -20.7 | -9.7 | -2.5 | -18.5 | -14.3 | -5.4 | -3 | **4.9** | -6.2 | **4.9** |
| Private Eye | 69571.3 | 24.9 | 72.8 | 59.6 | 81.9 | 24.8 | 86 | 93.9 | 100 | 46.3 | **110** |
| Qbert | 13455 | 163.9 | 1773.5 | 552.5 | 727 | 934.2 | 866.3 | 3620.1 | 3410.4 | 2760.3 | **3925** |
| Road Runner | 7845 | 11.5 | 11843.4 | 2606.4 | 5006.1 | 8724.7 | 12213.1 | **13429.4** | 12049.7 | 11892 | 12704.5 |
| Seaquest | 42054.7 | 68.4 | 304.6 | 272.9 | 315.2 | 310.5 | 558.1 | 532.9 | 628.3 | 600 | **653.7** |
| Up N Down | 11693.2 | 533.4 | 3075 | 2331.7 | 2646.4 | 3619.1 | **10859.2** | 10225.2 | 6675.7 | 6861 | 8602.5 |
| Interquartile Mean | 1.000 | 0.000 | 0.183 | 0.117 | 0.113 | 0.224 | 0.337 | 0.374 | 0.432 | 0.321 | **0.501** |
| Optimality Gap | 0.000 | 1.000 | 0.698 | 0.819 | 0.768 | 0.692 | 0.577 | 0.558 | 0.522 | 0.593 | **0.488** |

Table 4: Results on Atari-100k.

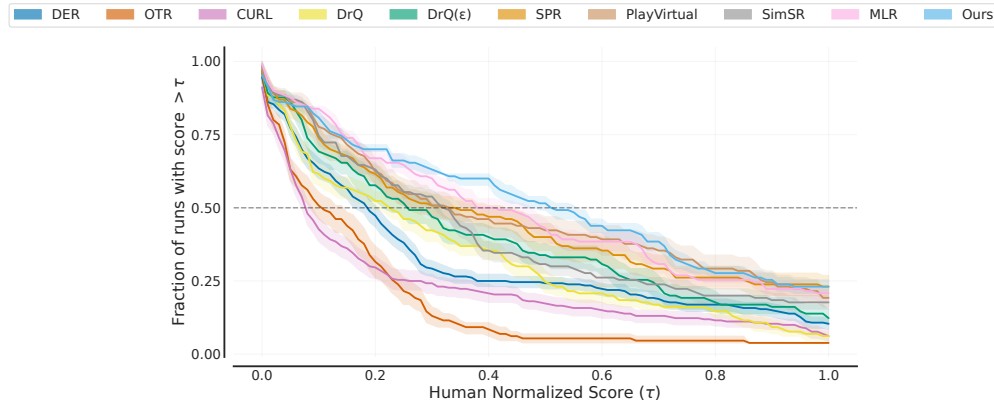

Figure 5: Performance profiles on the Atari-100k benchmark based on human-normalized score distributions. Shaded regions indicate 95% confidence bands.

difficult environments with sparse rewards, such as *Cartpole, Swingup Sparse* and *Hopper, Hop*, suggesting that our proposed model can effectively handle the collapse of the bisimulation objective function when the rewards are nearly zero. Additionally, in the *Reacher, Easy*, while SimSR quickly achieved high performance in the early stages of training, its performance gradually deteriorated, and variance increased over training, indicating that the dynamics model learned by SimSR may violate the compactness requirement, making it vulnerable to environmental noise and thus causing measurement expansion. In comparison, ReBis improved its performance significantly and rapidly, demonstrating its remarkable stability, which proves the superiority of implicit dynamics modeling.

## D.3 ABLATION STUDY

This section evaluates the impact of mask ratio, different objective components and hyperparameter design. We explored the impact of fixed learning objectives across varying mask ratios. We also examined the final results with a fixed mask ratio while learning the two objectives separately. Additionally, we adjusted the proportion of the two components under different mask ratios. We have conducted an ablation experiment focusing on the momentum coefficient $m$. These four ablation experiments were pivotal in dissecting the complex interactions between our learning objectives

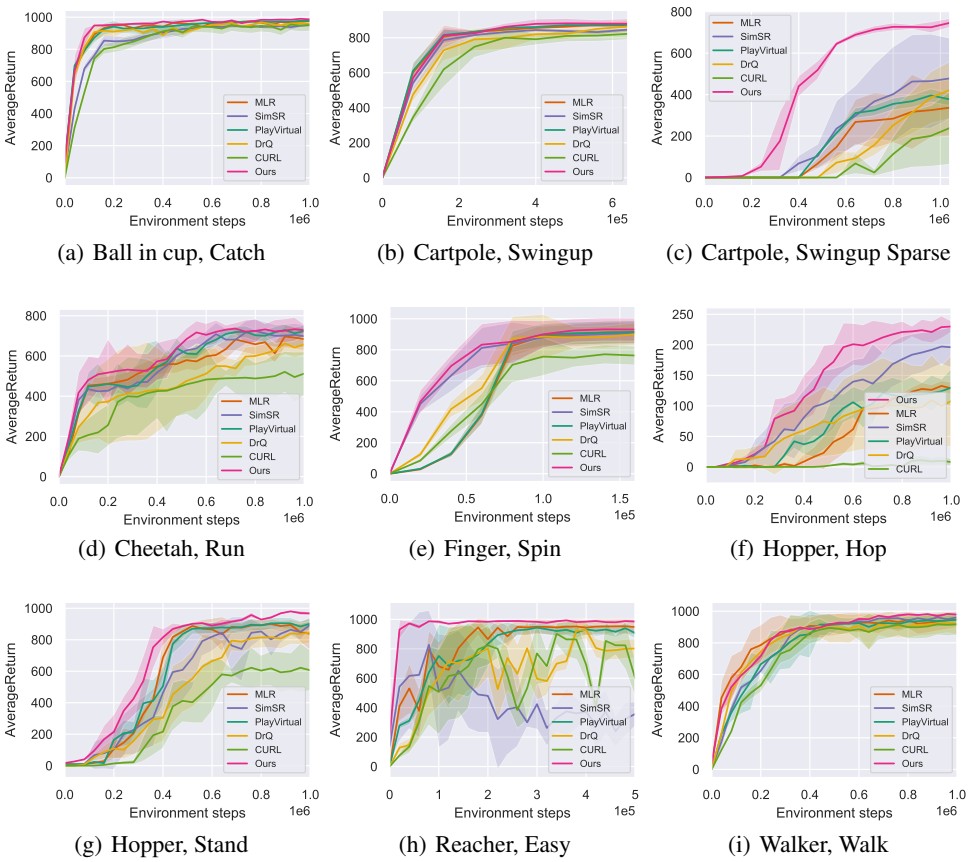

Figure 6: Results on DMContorl

and mask input. They offered insights into these factors' individual and combined effects on model performance. We conducted ablation studies on DMControl-500k and ran each model with 3 random seeds individually.

**Mask ratio.** We examined the effects of different mask ratios across four different tasks, running each model with 3 random seeds. Figure 7 shows the averaged evaluation results. Notably, contrary to the prevalent belief that images and videos contain a substantial amount of redundant information, our findings suggest that the optimal mask ratio for sequential tasks is approximately 0.5. This is relatively lower than the results in conventional vision domains (He et al., 2022; Feichtenhofer et al., 2022), which is nearly 0.9, yet it aligns with the result in Yu et al. (2022). We attribute this to the

| DMControl-100k | CURL | DrQ | PlayVirtual | MLR | SimSR | Ours |
|---|---|---|---|---|---|---|
| Ball in cup, Catch | 802±33 | 922±44 | 927±38 | 932±9 | 854±34 | **952±10** |
| Cartpole, Swingup | 619±101 | 728±59 | **817±33** | 809±56 | 792±43 | 812±46 |
| Cartpole, Swingup Sparse | 0±0 | 0±0 | 0±0 | 0±0 | 0±0 | **7±5** |
| Cheetah, Run | 210±80 | 302±123 | 448±73 | 453±36 | 435±35 | **481±117** |
| Finger, Spin | 756±123 | 885±144 | 897±73 | 890±71 | 894±89 | **904±94** |
| Finger, Turn Easy | 9±4 | 31±9 | 153±39 | 173±56 | 207±4 | **239±18** |
| Finger, Turn Hard | 0±0 | 0±0 | 3±3 | 8±4 | 9±5 | **13±2** |
| Hopper, Hop | 0±0 | 12±17 | 2±0 | 3±3 | 8±5 | **13±6** |
| Hopper, Stand | 15±2 | 86±42 | 24±3 | 34±13 | 81±99 | **168±88** |
| Pendulum, Swingup | 0±0 | 0±0 | 29±6 | 71±10 | 76±25 | **80±10** |
| Walker, Walk | 423±35 | 612±50 | 465±113 | **647±105** | 518±24 | 599±15 |

Table 5: Results (mean ± std) on the DMControl-100k benchmarks.

fact that an excessively low mask ratio may fail to filter out irrelevant spatiotemporal redundancy, while an overly high ratio may inadvertently remove valuable control-centric information. Moreover, considering that *Cartpole Swingup Sparse* and *Hopper Hop* are notably more challenging tasks than *Walker Walk* and *Reacher Easy*, it can be inferred that an appropriate mask ratio becomes increasingly vital for state representation learning as the task's difficulty escalates. Therefore, we adopt a mask ratio of 0.5 for all experiments.

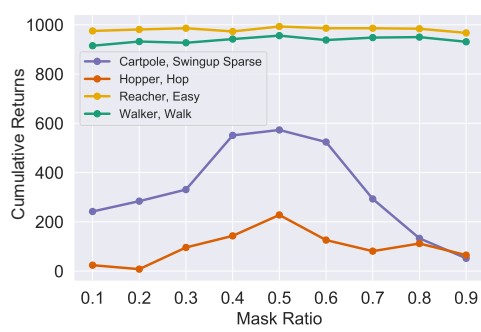

Figure 7: Comparison of different mask ratios in 4 different environments.

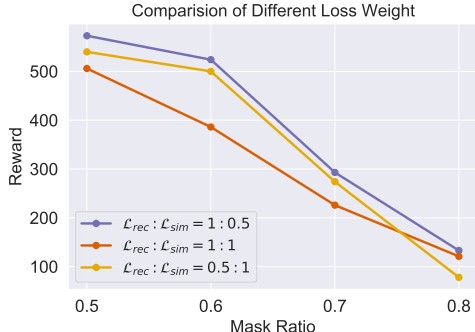

Figure 8: Comparison of different loss weight.

**Objective components.** We studied the contributions of different learning objective by selectively removing one to evaluate the resulting performance of the other. As shown in Table 6, the ablated objectives underperform compared to the complete objective, indicating that both components, $\mathcal{L}_{rec}$ and $\mathcal{L}_{sim}$, play crucial roles in our framework. It is worth noting that $\mathcal{L}_{rec}$ can be viewed as a simplified modification of MLR, omitting the projection and prediction in the head; hence, a marginally lower performance relative to MLR is expected. However, when combined with bisimulation-based components, the final performance experiences a substantial enhancement. This further demonstrates the effectiveness of our proposed framework.

| DMControl-500k | only $\mathcal{L}_{rec}$ | only $\mathcal{L}_{sim}$ | ReBis |
|---|---|---|---|
| Ball in cup, Catch | 953±11 | 946±23 | **984±2** |
| Cartpole, Swingup | 857±16 | 836±24 | **883±26** |
| Cart. Swing. Sparse | 25±7 | 86±17 | **518±45** |
| Cheetah, Run | 651±12 | 713±18 | **748±44** |
| Finger, Spin | 887±55 | 910±31 | **971±26** |
| Hopper, Hop | 97±31 | 203±22 | **234±13** |
| Hopper, Stand | 848±35 | 773±102 | **927±18** |
| Reacher, Easy | 869±24 | 876±33 | **992±4** |
| Pendulum, Swingup | 355±103 | 396±27 | **458±7** |
| Walker, Walk | 902±30 | 925±9 | **951±18** |

.

Table 6: Comparison of different objective components

**Weighing the losses.** We aim to find the optimal balance between reconstruction loss and bisimulation loss under varying high mask ratios to achieve the best performance. The proportion of loss can be flexibly designed. We aim to find the optimal balance between reconstruction loss and bisimulation loss under varying high mask ratios to achieve the best performance. As shown in Figure 8, we tried 3 different proportion schemes in *Cartpole, Swingup Sparse* environment. According to the conducted experiments, the optimal performance is achieved when the ratio of reconstruction loss to bisimulation loss is set to 1:0.5. This indicates the significant role that reconstruction loss plays in the model training process. The reconstruction loss helps the model to better capture reward-free control information.

**Momentum coefficient.** In this section, we present an in-depth ablation study focused on assessing the impact of momentum coefficient $m$ variations on the performance of our model. Our investigation specifically explored three distinct values of the momentum coefficient: 0.9, 0.95, and 0.99. These values were chosen to represent a range from moderate to high momentum,

| momentum $m$ | 0.9 | 0.95 | 0.99 |
|---|---|---|---|
| Finger, Turn Easy | 596±6 | 652±34 | 508±12 |
| Pendulum, Swingup | 440±11 | 458±7 | 403±6 |

.

Table 7: Comparison of different momentum $m$

allowing us to observe how different levels of 'memory' in the algorithm affect its performance in various environmental settings. The experimental results, as summarized in the accompanying table 7, provide a comparative analysis of the model's performance across these momentum settings in different environments.

## E   IMPACT OF LATENT SPACE RECONSTRUCTION ON MODEL LEARNING

We propose the following discussion which aims at explaining how the process of reconstruction in the latent space affects the learning of our model ReBis. In a departure from conventional methodologies, our approach integrates a unique blend of the bisimulation loss for capturing reward-specific information and the reconstruction loss in the latent space for eliminating spatiotemporal redundancies.

Reconstructing a visual signal with high information density through low-dimensional feature embeddings, a common practice in the computer vision domain, can successfully preserve spatiotemporal information. However, it is unnecessary and inefficient to reconstruct at the pixel level as this contains significant redundancies. Therefore, we opt to reconstruct the latent features instead of raw observations, maintaining essential information relevant to control while reducing unnecessary spatiotemporal redundancies. The incorporation of a masking strategy further streamlines the reconstruction process within the latent space.

It is essential to recognize that both components assume pivotal roles, underscoring the notion that the control-centric representation is not contingent upon a single element. However, it is noteworthy that, given the nature of the bisimulation metric, it primarily excels at learning reward-specific information rather than control-centric information. In environments characterized by sparse rewards, as stipulated in Theorem 4, bisimulation inevitably encounters limitations. The incorporation of our latent space reconstruction loss aims to mitigate this inherent limitation, as elucidated in Theorem 5. In the case where the reconstruction loss is used, even within such challenging environments, agents can still engage with their surroundings, gaining insights into self-control strategies. This premise underscores the primary motivation behind our work.

## F   IMPLEMENTATION DETAILS

### F.1   HYPERPARAMETERS

We present all hyperparameters used for the Atari-100k benchmark in Table 8 and the DMControl benchmarks in Table 9. We follow prior work (i.e., SAC (Haarnoja et al., 2018) and Rainbow (Hessel et al., 2018)) for the policy learning hyperparameters.

### F.2   NETWORK ARCHITECTURE

Our structure is based on an transformer encoder. This transformer encoder comprises two standard attention layers, each have a single attention head. To ensure compatibility in dimensions between the state and action representations, we employ an FC layer as the action embedding head. Besides, we utilize relative position embeddings in our model, drawing inspiration from the prevailing techniques in the domain of transformer architectures (Vaswani et al., 2017). By doing so, we adopt the relative positional embeddings to encode relative temporal positional information into both state and action tokens.

### F.3   ALGORITHM

The training algorithm of our method is presented in Algorithm 1.

### F.4   HARDWARE AND WALL-CLOCK TIME

We implemented ReBis and performed all experiments with the PyTorch framework, based on a server with 4 GeForce RTX 3090 GPUs, 48 Intel Xeon CPUs, and 128GB memory. On Atari, the average wall-clock training time is 8.1 and 8.6 hours for ReBis and MLR respectively. On

---

**Algorithm 1** ReBis

---

**Require**: The online encoder $\phi$, the momentum encoder $\hat{\phi}$, the Transformer $G$, and the policy learning networks $\omega$, parameterized by $\theta_\phi$, $\theta_{\hat{\phi}}$, $\theta_G$, and $\theta_\omega$, respectively; the cube masking function CubeMask($\cdot$), the optimizer Optimizer($\cdot$, $\cdot$).

1: Determine the weight of bisimulation loss $\beta$, observation sequence length $K$, mask ratio $\eta$, cube shape $k \times h \times w$, and EMA coefficient $m$.
2: Initialize a replay buffer $\mathcal{D}$.
3: Initialize all parameters.
4: **while** train **do**
5:     Interact with the environment based on the policy
6:     Sample the sequence of $K$ observations $\{o_t, o_{t+1}, \ldots, o_{t+K-1}\}$ and corresponding actions $\{a_t, a_{t+1}, \ldots, a_{t+K-1}\}$ from replay buffer $\mathcal{D}$
7:     Cube masking the observation sequence:
    $\{o'_t, o'_{t+1}, \ldots, o'_{t+K-1}\} \leftarrow CubeMask(\{o_t, o_{t+1}, \ldots, o_{t+K-1}\})$
8:     Siamese Encoding:
    $\tau_K^s \leftarrow \{\phi(o'_t), \phi(o'_{t+1}), \ldots, \phi(o'_{t+K-1})\}$,
    $\tau_K^{\hat{\phi}(o)} \leftarrow \{\hat{\phi}(o_t), \hat{\phi}(o_{t+1}), \ldots, \hat{\phi}(o_{t+K-1})\}$
9:     Perform dynamics model: $\hat{\tau}_K^s \leftarrow G(\tau_K^s)$
10:     Calculate reconstruction loss: $\mathcal{L}_{\text{rec}} = \text{MSE}(\hat{\tau}_K^s, \tau_K^{\hat{\phi}(o)})$
11:     Calculate bisimulation loss: $\mathcal{L}_{\text{sim}} = \text{MSE}\left(\bar{d}(\hat{\phi}(o_i), \hat{\phi}(o_j)), \mathcal{F}^\pi \bar{d}(\hat{\phi}(o_i), \hat{\phi}(o_j))\right)$
12:     Updating with joint objective: $\mathcal{L}_{\text{total}} = \mathcal{L}_{\text{rec}} + \beta \mathcal{L}_{\text{sim}}$
13:     Calculate RL loss $\mathcal{L}_{\text{RL}}$ based on a given base RL algorithm (e.g., SAC)
14:     Update online parameters: $\theta_\phi, \theta_{\hat{\phi}}, \theta_G, \theta_\omega \leftarrow Optimizer(\theta_\phi, \theta_{\hat{\phi}}, \theta_G, \theta_\omega, (\mathcal{L}_{\text{RL}}, \mathcal{L}_{\text{total}}))$
15:     Update momentum parameters: $\hat{\phi} \leftarrow m\hat{\phi} + (1 - m)\phi$
16: **end while**

---

DM-Control tasks, the averaged wall-clock training time is 6.2 and 6.4 hours for ReBis and MLR, respectively.

One can observe that the training time of ReBis is marginally faster than the MLR. This disparity is possibly due to the complexity of MLR's model structure. In contrast to ReBis, MLR employs additional projection and prediction heads, as well as a momentum projection head specifically for reconstruction targets. These additional layers contribute to increased computational demand for forward pass on each iteration during training, which consequently extends the overall training duration. In summary, our proposed method, ReBis, can expedite the training process while ensuring superior outcomes, compared with MLR.

| Hyperparameter | Value |
| --- | --- |
| Stacked frames | 4 |
| Observation shape | (84, 84) |
| Action repeat | 4 |
| Replay buffer size | 100000 |
| Training steps | 100000 |
| Max frames per episode | 108000 |
| Minimum sampling replay size | 2000 |
| Mini-batch size | 64 |
| Optimizer | Adam |
| Optimizer: learning rate | 0.0001 |
| Optimize: $\beta_1$, $\beta_2$ | 0.9,0.999 |
| Optimize: $\epsilon$ | 0.00015 |
| Encoder EMA $m$ | 0.95 |
| Latent dimension $d$ | 50 |
| Max gradient norm | 10 |
| Update | Distributional Q |
| Dueling | True |
| Support of Q-distribution | 51 bins |
| Discount factor | 0.99 |
| Reward clipping Frame stack | [-1, 1] |
| Priority exponent | 0.5 |
| Priority correction | $0.4 \rightarrow 1$ |
| Exploration | Noisy nets |
| Noisy nets parameter | 0.5 |
| Evaluation trajectories | 100 |
| Replay period every step | 1 |
| Updates per step | 2 |
| Multi-step return length | 10 |
| Q network: channels | 32, 64, 64 |
| Q network: filter size | $8\times8, 4 \times 4, 3 \times 3$ |
| Q network: stride | 4, 2, 1 |
| Q network: hidden units | 256 |
| Weight of bisimulation loss $\beta$ | 0.5 |
| Mask ratio $\eta$ | 50% |
| Sequence length $K$ | 16 |
| Cube spatial size $h \times w$ | $12 \times 12$ |
| Cube depth $k$ | 8 |

Table 8: Hyperparameters used for Atari benchmark.

| Hyperparameter | Value |
| --- | --- |
| Stacked frames | 3 |
| Observation shape | (84, 84) |
| Replay buffer size | 100000 |
| Initial steps | 1000 |
| Action repeat | 2 *Finger-spin* and *Walker-walk*; |
| | 8 *Cartpole-swingup*; |
| | 4 *otherwise* |
| Evaluation episodes | 10 |
| Optimizer | Adam |
| $(\beta_1, \beta_2) \rightarrow (\theta_\phi, \theta_G, \theta_\omega)$ | (0.9,0.999) |
| $(\beta_1, \beta_2) \rightarrow (\alpha)$ | (0.5,0.999) |
| Learning rate | 0.0002 *Cheetah-run*; |
| | 0.001 *otherwise* |
| Batch size | 128 |
| Q-function EMA $m$ | 0.99 |
| Critic target update freq | 2 |
| Discount factor $\gamma$ | 0.99 |
| Encoder EMA $m$ | 0.9 *Walker-walk*; |
| | 0.95 *otherwise* |
| Latent dimension $d$ | 50 |
| Weight of bisimulation loss $\beta$ | 0.5 |
| Mask ratio $\eta$ | 50% |
| Sequence length $K$ | 16 |
| Cube spatial size $h \times w$ | $10 \times 10$ |
| Cube depth $k$ | 4 *Cartpole-swingup* and *Reacher-easy*; |
| | 8 *otherwise* |

Table 9: Hyperparameters used for DMControl benchmark.

