# OpenReview forum: "Towards Control-Centric Representations in Reinforcement Learning from Images"
_ICLR.cc/2024/Conference — Submitted to ICLR 2024_

### Official Review · Reviewer_sEtP · 2023-10-30

**Soundness:** 2 fair
**Presentation:** 2 fair
**Contribution:** 2 fair
**Rating:** 5
**Confidence:** 3

**Summary:**

The authors propose to improve the bisimulation principle methods in terms of (1) better latent dynamics prediction; (2) handle sparse reward environments. The authors propose ReBis, which combine bisimulation loss with a masked transformer model to learn the latent forward dynamics and latent reconstruction loss can prevent collapse from uninformative rewards.

**Strengths:**

The motivation is clear. The paper identifies the drawbacks of the existing bisimulation methods, and proposes corresponding algorithms to handle them. In general, the writing is clear and the paper is easy to follow.

**Weaknesses:**

1. The novelty remains concerned. It seems that the algorithm combines MLR + a bisimulation loss.

2. The algorithm designs (using masked inputs, momentum encoder, etc.) need ablation studies to prove its effectiveness.

3. Though the motivation is to improve the bisimulation method, the proposed algorithm seems to be something between model-based RL (as cited in “Highly Expressive Dynamics Model” part). So I think it is necessary to discuss / compare with these works.

**Questions:**

1. I find some notations might need more clarification:

a) By stacking 3 frames for each observation (derive $o_t ‘$ from $o_t$), do you stack 3 frames $(o_t, o_t, o_t)$ with different masking, stack $(o_t, o_{t+1}, o_{t+2})$, or other setting?

b) Can you please explain the relative position embedding $\tau_K^p$ in detail? Do you add this position embedding to the state / action tokens, or concatenate this position embedding as an independent input?

c) As for the block masking, I think it will be better to introduce how the images are masked to make the paper self-contained.

2. What are the modifications on the proposed methods w.r.t. MLR besides the behavior loss (bisimulation part)? Most of the merits of the proposed methods (e.g., better forward dynamics, learn asymmetric reconstruction to handle uninformative reward) seems to come from the MLR backbone.

3. I’m also wondering why using the MLR as backbone to improve bisimulation and why not other methods (e.g., Dreamer, TransDreamer, Transformer world model) as cited in your related works.

---

> ### Author Response · Authors · 2023-11-17
> **Response to Reviewer sEtP - 1**
>
> Thanks for your comments, we would like to clarify some points and explain your questions.
>
> **1. Explanation of frame stack.**
>
> We use the commonly used frame stack method, which involves stacking $(o_t, o_{t+1}, o_{t+2})$. This helps capture information about object motion, velocity, and direction,  favoring the tasks with an underlying POMDP.
>
> **2. Explanation of the relative position embedding.**
>
> We did utilize relative position embeddings in our model, drawing inspiration from the prevailing techniques in the domain of transformer architectures. By doing so, we adopt the relative positional embeddings to encode relative temporal positional information into both state and action tokens. We will consider offering a more detailed description in subsequent revisions.
>
> **3. Explanation of the Block-wise masking.**
>
> The block masking technique we employed is inspired by the method described in [1], which utilizes both spatial and temporal masking blocks. We divide the sequence of image frames into several cubic segments. This segmentation considers both spatial and temporal dimensions, ensuring that each cube encompasses a portion of an image as well as its temporal extension over consecutive frames. The cubes are selected by initially defining a 2-D block at a random time step in the sequence of frames. Once a 2-D block is defined, it is extended along the temporal dimension. For the actual masking process, we randomly select these cubes until we achieve a predetermined masking ratio. Recognizing the importance of making the paper self-contained, we will provide a more detailed description in the future version of our paper.
>
> [1] Wei, C., Fan, H., Xie, S., Wu, C. Y., Yuille, A., & Feichtenhofer, C. (2022). Masked feature prediction for self-supervised visual pre-training. In Proceedings of the IEEE/CVF Conference on Computer Vision and Pattern Recognition (pp. 14668-14678).
>
> **4. The key difference between MLR and the proposed method ReBis.**
>
> We have provided a detailed analysis in Appendix C, comparing the ReBis with existing methods, particularly MLR. Firstly, it is essential to clarify that while ReBis shares some similarities with MLR, such as the adoption of a mask strategy and a transformer-based encoder, these elements are not unique to MLR and are widely used in current models. The key to differentiate a model is how it defines its objectives and training process when applying these frameworks. In contrast to MLR, ReBis is fundamentally based on the principles of bisimulation. This influences our method to tailor its objectives and training process specifically to enhance bisimulation. As a result, ReBis achieves an effective balance between efficiency and capability, particularly in capturing control-centric (reward-associated) state dynamics. This is a significant departure from MLR, which tends to overlook task-related information.
>
> From the model architecture perspective, MLR employs a more complex network structure, including additional two projection heads, and one prediction head. While these components result in higher complexity and computational demands and are integral to MLR, ReBis does not require such complex structures to effectively perform its tasks.
>
> Therefore, both in terms of design target and model architecture, ReBis significantly diverges from MLR, showcasing its unique approach and contribution to the field.
>
> **5. Why not other methods (e.g., Dreamer)?**
>
> We did consider and experiment with structures like Dreamer to enhance bisimulation performance. However, we encountered several issues. Firstly, methods in the Dreamer series are model-based, which inherently differs from the approach we aimed for in our research. Secondly, Dreamer involves pixel-level reconstruction.  Our goal was to minimize irrelevant information as much as possible.  Therefore, the reconstruction nature of Dreamer essentially conflicts with the objectives of bisimulation, which seeks to abstract important information away from such low-level details.
>
> Given these considerations, it is currently not feasible to apply bisimulation to methods like Dreamer. On the other hand, MLR, with its focus on latent space reconstruction, aligns more closely with the principles of bisimulation. Furthermore, it is indeed possible that there are other approaches that could potentially yield equal or better results than ReBis. Our focus has been on establishing a paradigm in which certain modules are interchangeable. We plan to explore the potential of integrating ReBis with other methods in future work.

---

> ### Author Response · Authors · 2023-11-17
> **Response to Reviewer sEtP - 2**
>
> **6. Ablation studies.**
>
> Thank you for your comment regarding the need for ablation studies to validate the effectiveness of the algorithm design, particularly the use of masked inputs and momentum encoder. Masking inputs is a fundamental step in our methodology, as it is integral to the subsequent reconstruction process, which plays a crucial role in our model's learning mechanism. We would like to emphasize that to explore its effectiveness, we have conducted thorough ablation experiments focusing on different masking ratios. The detailed results and analyses of these ablation studies are presented in Appendix D.3 of our paper.
>
> The momentum update technique has become a common practice due to its effectiveness in enhancing the stability of the learning process. Our inspiration to use the momentum encoder for representation learning comes from its successful adaptation in references [2, 3, 4]. In response to your suggestion, we have conducted an ablation experiment focusing on the momentum coefficient $m$. The results of this experiment are as follows:
>
> |    momentum $m$     | 0.9  |      | 0.95 |      | 0.99 |      |
> | ----------------- | ---- | ---- | ---- | ---- | ---- | ---- |
> |                   | mean | std  | mean | std  | mean | std  |
> | Finger, Turn Easy | 596  |  6   | 652  | 34   | 508  | 12   |
> | Pendulum, Swingup | 440  |  11  | 458  | 7    | 403  | 6    |
>
> [2] He, K., Fan, H., Wu, Y., Xie, S., & Girshick, R. (2020). Momentum contrast for unsupervised visual representation learning. In Proceedings of the IEEE/CVF conference on computer vision and pattern recognition (pp. 9729-9738).
>
> [3] Laskin, M., Srinivas, A., & Abbeel, P. (2020, November). Curl: Contrastive unsupervised representations for reinforcement learning. In International Conference on Machine Learning (pp. 5639-5650). PMLR.
>
> [4] Zang, H., Li, X., & Wang, M. (2022, June). Simsr: Simple distance-based state representations for deep reinforcement learning. In Proceedings of the AAAI Conference on Artificial Intelligence (Vol. 36, No. 8, pp. 8997-9005).
>
> **7. Comparison with model-based RL method.**
>
> Indeed, ReBis is a model-free method, as we does not construct the reward function and the explicit reconstruction in observation space. In our framework, we only reconstruct the state embeddings in latent space, which means categorizing our method into model-based RL seems not to be appropriate. Nevertheless, we still constructed an experiment on comparing our method and a model-based representation method TIA [5] (one method that incorporates Dreamer structure and also aims to learn task-specific information), to show the effectiveness of our model:
>
> |                    |  TIA |      | ReBis |      |
> | ------------------ | ---- | ---- | ---- | ---- |
> |                    | mean | std  | mean | std |
> | Ball in cup, Catch | 750  |  401 | 970  | 16 |
> | Cartpole, Swingup  | 108  |  18  | 859  | 21 |
> | Finger, Spin       | 485  |  27  | 893  | 35 |
> | Hopper, Hop        | 57   |  67  | 145  | 28 |
>
> Please note that due to the tight rebuttal timeframe, these experimental results are preliminary. We plan to include more model-based methods into comparison in subsequent versions, to further demonstrate the advantages of our model.
>
> [5] Fu, X., Yang, G., Agrawal, P., & Jaakkola, T. (2021, July). Learning task informed abstractions. In International Conference on Machine Learning (pp. 3480-3491). PMLR.

---

> > ### Comment · Reviewer_sEtP · 2023-11-21
> >
> > Thank you for your response, I appreciate your response and I would like to raise the score. However, the major concern w.r.t. the novelty remains (W1). According to the author's feedback, the main difference is that: (1) ReBis basically use the MLR network but removes "additional two projection heads, and one prediction head", and (2) a bisimulation training objective. To me, this still seems like a combination of two existing methods as I commented previously, and this is why I don't raise my score further

---

> > > ### Author Response · Authors · 2023-11-21
> > > **Response to Reviewer sEtP - 3**
> > >
> > > We are grateful for the reviewer's valuable suggestions and the positive response to our replies, which have significantly contributed to improving our paper's quality. However, we respectfully disagree with the reviewer's assessment of the novelty in our work.
> > >
> > > **1. Removing projection heads and prediction head is non-trivial.**
> > >
> > > The removal of projection heads and the prediction head from the MLR[1] structure is not as straightforward as it might seem. While the mask strategy and transformer-based encoder in MLR are commonly used and not unique, the model's distinction lies in its additional two projection heads and one prediction head. This design, closely following BYOL[2]'s framework and serving a role similar to contrastive learning[3], is crucial for MLR's functionality, as highlighted in Appendix C of the MLR paper. The projection and prediction heads, along with the MLR-type similarity loss, are vital components; their absence leads to markedly poorer performance, as evidenced by Table 7 in the MLR paper. Moreover, our manuscript discusses how, despite its structure and contrastive-like loss, the MLR agent struggles to capture task-specific knowledge, particularly reward-related information. This causes its inability to handle environments rich in spatiotemporal redundancy, a limitation evident in most of our experiments. In contrast, our approach, grounded in the bisimulation principle, not only addresses spatiotemporal redundancy but also effectively learns task-specific information.
> > >
> > > **2. Making the bisimulation principle work well is not simple.**
> > >
> > > Our main contribution is the development of an effective approach tailored to addressing the shortcomings of conventional bisimulation methods.  Our method stands out as the first to integrate a transformer structure for dynamics modeling within the bisimulation framework. Additionally, it is the inaugural bisimulation-based method to achieve state-of-the-art performance in concurrent studies. While previous efforts like DBC[4], MICo[5], SimSR[6], and others have shown comparable performance to other representation methods in certain tasks, they generally do not match state-of-the-art methods in a wide range of tasks. We believe our work represents a significant advancement in exploring the bisimulation principle. This development is likely to spark heightened interest and stimulate more in-depth research into bisimulation in community.
> > >
> > > **3. The innovation of a framework should not be judged solely by the novelty of its individual modules.**
> > >
> > > For instance, TD3BC[7] merely adds BC regularization and state normalization to the TD3 algorithm, while BYOL only introduces an extra prediction head. Despite these seemingly minor additions, both frameworks are well-founded, effectively demonstrating the logic and necessity of their respective models. Similarly, while some modules in our approach are also commonly used, our methodology is rooted in thorough theoretical analysis and empirical evidence. We have extensively justified the need for our model both theoretically and empirically. Thus, our approach is driven and substantiated by theory.
> > >
> > > Indeed, we commend and respect the reviewers' cautious stance on paper acceptance, recognizing its importance in upholding ICLR's high standards and impact. Nevertheless, in light of our detailed responses, we hope for a re-evaluation of our paper's quality. We also welcome any differing opinions and are fully prepared to address further queries or implement additional suggestions.
> > >
> > >
> > > [1] Tao Yu, Zhizheng Zhang, Cuiling Lan, Yan Lu, Zhibo Chen: Mask-based Latent Reconstruction for Reinforcement Learning. NeurIPS 2022
> > >
> > > [2] Jean-Bastien Grill, Florian Strub, et al.: Bootstrap Your Own Latent - A New Approach to Self-Supervised Learning. NeurIPS 2020
> > >
> > > [3] Yuandong Tian, Xinlei Chen, Surya Ganguli: Understanding self-supervised learning dynamics without contrastive pairs. ICML 2021: 10268-10278
> > >
> > > [4] Zhang, A., McAllister, R., Calandra, R., Gal, Y., & Levine, S. (2020). Learning invariant representations for reinforcement learning without reconstruction. arXiv preprint arXiv:2006.10742.
> > >
> > > [5] Castro, P. S., Kastner, T., Panangaden, P., & Rowland, M. (2021). MICo: Improved representations via sampling-based state similarity for Markov decision processes. Advances in Neural Information Processing Systems, 34, 30113-30126.
> > >
> > > [6] Zang, H., Li, X., & Wang, M. (2022, June). Simsr: Simple distance-based state representations for deep reinforcement learning. In Proceedings of the AAAI Conference on Artificial Intelligence (Vol. 36, No. 8, pp. 8997-9005).
> > >
> > > [7] Scott Fujimoto, Shixiang Shane Gu: A Minimalist Approach to Offline Reinforcement Learning. NeurIPS 2021: 20132-20145

---

### Official Review · Reviewer_6U7G · 2023-10-30

**Soundness:** 3 good
**Presentation:** 4 excellent
**Contribution:** 3 good
**Rating:** 6
**Confidence:** 4

**Summary:**

This paper builds on the idea of applying the bisimulation principle to shape the representation of deep RL with task-specific information. The authors first identify several limitations of prior work, for example, the expressiveness of the learned dynamics model and the ability to leverage reward-free control information when the reward is uninformative.

Then, the authors propose ReBis which leverages spatiotemporal consistency and long-term behavior similarity. More specifically, they reduce spatio-temporal redundancy in observations via Siamese encoders with block-wise masking and use a transformer-based dynamics model to capture multi-modal behaviors.

Finally, the authors demonstrate the superiority of ReBis on Atari and DeepMind control suites.

**Strengths:**

This paper compares with many prior works on learning a good representation for policy learning, including CURL, SimSR, etc., and shows their proposed method, ReBis, outperforms them in most tasks. Moreover, their learned representation is more robust to different backgrounds as demonstrated in the DeepMind control suite with different background distractions.

Experiments and ablations are comprehensive and well-designed. The ablation of different loss components justifies the design of ReBis and shows both reconstruction loss and bi-simulation loss contribute to the gain in ReBis.

The paper writing is easy to follow and has a nice overview of related work.

**Weaknesses:**

As pointed out by the author, the representation learning part adds non-trivial complexity to the RL algorithm. It would be interesting to experiment with different update frequencies of the representation learning part. For example, what’s the score if we update the representation encoder every 1, 10, 100, or 1000 environmental interactions?

Is the method complementary to other representation learning methods, such as augmentation, video prediction, time contrastive learning, etc? Does combining with other methods lead to better results?

The representation learning part relies on a reward specification by design, which prevents it from being used as a general pre-training method with unknown tasks. The authors should consider discussing this in limitations.

**Questions:**

Do we need to update the encoder every RL update? Can we update it less frequently than the policy?

Is the method complementary to other representation learning approaches?

Can we use it for pre-training a representation without task specification?

---

> ### Author Response · Authors · 2023-11-17
> **Response to Reviewer 6U7G**
>
> Thanks for your comments, we would like to clarify some points and explain your questions.
>
> **1. Do we need to update the encoder every RL update? Can we update it less frequently than the policy?**
>
> Our current method does indeed require updating the encoder at every RL update. This frequent updating is integral to ensuring that the encoder accurately captures the evolving state representations, which is crucial for the effectiveness of our RL algorithm. However, inspired by your suggestion, we explored the possibility of updating the representation encoder less frequently.
>
> To test this, we conducted experiments where the encoder was updated every 10, 100, or 1000 environmental interactions. The results of these experiments are as follows:
> |                          |  10  |      | 100  |      | 1000 |      |
> | ------------------------ | ---- | ---- | ---- | ---- | ---- | ---- |
> |                          | mean | std  | mean | std  | mean | std  |
> | Cartpole, Swingup Sparse | 492  | 8    | 290  | 46   | 141  | 53   |
> | Pendulum, Swingup        | 290  | 28   | 110  | 47   | 22   | 16   |
>
> These findings suggest that while our current approach of updating the encoder at every RL update is effective, there may be potential in exploring alternative update frequencies. The need for frequent updates stems from the dynamic nature of reinforcement learning environments. Updating the encoder with every RL update ensures that the state representations remain current and relevant, thereby facilitating more effective learning and decision-making. Less frequent updates may lead to a decreased ability of the encoder to keep pace with the rapidly changing environment and agent dynamics. These findings open avenues for future research to optimize the balance between update frequency and computational efficiency, potentially through adaptive updating mechanisms or more efficient methods.
>
> **2. Is the method complementary to other representation learning approaches?**
>
> Our method can be effectively combined with data augmentation techniques, such as random crop.  Prior research [1, 2] has uncovered that applying proper data augmentation can significantly improve the sample efficiency of reinforcement learning models.
>
> The integration of video prediction techniques with our method may offer additional contextual information, aiding in a better understanding of temporal dynamics. Similarly, time contrastive learning, which focuses on understanding the temporal relationships between different observations, could also complement our approach.  By leveraging the temporal contrasts, our method might further enhance its ability to discern meaningful patterns in temporal sequences.
>
> In our future work, we believe that incorporating these approaches could potentially lead to significant improvements in the model's performance, particularly in scenarios with complex temporal dependencies.
>
> [1] Laskin, M., Lee, K., Stooke, A., Pinto, L., Abbeel, P., & Srinivas, A. (2020). Reinforcement learning with augmented data. Advances in neural information processing systems, 33, 19884-19895.
>
> [2] Kostrikov, I., Yarats, D., & Fergus, R. (2020). Image augmentation is all you need: Regularizing deep reinforcement learning from pixels. arXiv preprint arXiv:2004.13649.
>
> **3. Can we use it for pre-training a representation without task specification?**
>
> Our model is primarily designed to extract task-specific features, which is central to its effectiveness in sparse reward environments.   This focus ensures that the model could capture relevant information that directly contributes to the task.
>
> Despite this task-specific orientation, we conducted exploratory experiments to assess the model's adaptability to different environments.   In one such experiment, we used an encoder trained in the 'Finger Spin' environment to evaluate its performance across 'Finger Turn Easy’ environment. The results of this experiment are as follows:
>
> | an encoder trained on Finger, Spin | 500k |      |
> | ---------------------------------- | ---- | ---- |
> |                                    | mean | std  |
> |       Finger, Turn Easy            | 611  | 21   |
>
> These findings suggest that the learned representation will generalize to unseen reward functions, it also exhibits a degree of flexibility and can adapt to new environments to some extent. Based on these results, it appears that our model can be used for pre-training without explicit task specifications. However, the effectiveness of this pre-training may vary depending on the similarity between the training and target environments.

---

> > ### Comment · Reviewer_6U7G · 2023-11-21
> >
> > Thank you for the new experiments! This partially addresses my concern. However, the fact that the encoder needs frequent updates and its lack of generalizability across tasks made me worry about its applicability in more complex environments. Thus, I decided to keep my original score!

---

> ### Author Response · Authors · 2023-11-22
> **Response to Reviewer 6U7G**
>
> We appreciate the reviewer’s feedback to our replies and the concerns about the  generalizability and applicability of our method.
>
> In terms of generalizability for in-domain tasks, our approach holds distinct advantages. These arise from our method's integration of both task-specific and spatiotemporal dependencies, a claim substantiated by our latest experiments mentioned in the Answer to Q3. These findings further demonstrate that the ReBis encoder maintains superior performance over current baselines, even after adopting the encoder from another in-domain task. Furthermore, we recognize that the reviewer's concern may arise from bisimulation theory's inherent focus on integrating behavioral similarity into representation learning, which naturally leads to task or domain specificity. Adhering to the 'no free lunch theorem', we understand that a model tailored for a specific task may not universally adapt to all environments, which is why our evaluations are concentrated solely on in-domain tasks. And models with higher generalizability with cross-domain tasks often necessitates more in-domain-specific tuning. Hence, the trade-off between generalizability and task-specific (or domain-specific) optimality is a direct consequence of the model's design principles.
>
> Additionally, it is important to note that all baseline approaches used for comparison in our study update their encoder with the same frequency as ours. This frequency setting is also consistent with the theoretical requirement of a high update frequency for the encoder, given the dependence of the bisimulation objective on the policy, and it will not hinder the effectiveness of our model in complex environments. Besides, this does not affect inference complexity as we only use CNN encoder during inference time.  Nevertheless, the time complexity deserves attention here, as the incorporation of a transformer architecture in our training process results in considerable time requirements for each gradient update. Therefore, future work could interestingly focus on accelerating model forward speed and achieving model distillation without compromising accuracy.

---

### Official Review · Reviewer_TC9E · 2023-11-01

**Soundness:** 4 excellent
**Presentation:** 3 good
**Contribution:** 3 good
**Rating:** 8
**Confidence:** 4

**Summary:**

The paper deals with image-based reinforcement learning. Image-based RL requires the extraction of control-specific relevant information from the images while discarding the visual noise. The authors came up with a representation learning algorithm that uses the bisimulation metric to measure distances between states. Bisimulation suffers from issues such as possible collapse in sparse rewards and requiring dynamics modeling which the authors are able to alleviate. The produce competitive results on Atari and distracting dm control suite.

**Strengths:**

(1) They correctly identify the issues in using the bisimulation metric for quantifying the distances between states. They take measures to solve these.

(2) They use a transformer to implicitly learn the dynamics which improves the expressibility of the dynamics models.

(3) Their reconstruction objective ensures that the representations do not collapse even in a sparse rewards regime.

(4) They produce impressive results in the distracting dm control suite.

**Weaknesses:**

(1) Can this loss be used as a pretraining loss for learning an encoder? There should have been some experiment depicting this.

(1) The method of representation learning seems way more complex than the RL algorithm itself. If this is an auxiliary loss, the transformer model capturing the dynamics is never used by the RL algorithm which seems a waste of resources.

**Questions:**

(1) What if the distracting images in distracting dm control are added during training as well? Can the model get distracted then?

---

> ### Author Response · Authors · 2023-11-17
> **Response to Reviewer TC9E**
>
> Thanks for your comments, we would like to clarify some points and explain your questions.
>
> **1. Can this loss be used as a pretraining loss for learning an encoder?**
>
> Thank you for your question. We have experimented with this possibility. We used an encoder pre-trained in one environment (Finger Spin) to evaluate its performance in different environment (Finger Turn Easy). The result of the experiment is as follows:
>
> | an encoder trained on Finger, Spin | 500k |      |
> | ---------------------------------- | ---- | ---- |
> |                                    | mean | std  |
> |       Finger, Turn Easy            | 611  | 21   |
>
> This experiment demonstrates that our method is versatile enough to be used for pre-training representations without task-specific constraints. The trained encoder shows promising adaptability and generalization across various environments, indicating its potential for broader applications in reinforcement learning.
>
> **2. The method of representation learning seems way more complex than the RL algorithm itself. If this is an auxiliary loss, the transformer model capturing the dynamics is never used by the RL algorithm which seems a waste of resources.**
>
> While the transformer model primarily focuses on capturing the dynamics for representation learning, its contribution extends beyond being just an auxiliary loss. The learned representations, albeit indirectly, play a vital role in the overall RL process. They provide the RL algorithm with a more nuanced understanding of the environment. Regarding the concern about resource utilization, Rebis aims to balance the computational overhead with the benefits brought by the transformer model. We understand the importance of efficient resource usage and continuously strive to optimize our model to make the best use of available resources. The current design reflects a trade-off where the added complexity is counterbalanced by the substantial improvements in learning quality.
>
> We acknowledge the need to explore ways to directly leverage the dynamics captured by the transformer model within the RL algorithm. Future iterations of our research will focus on more tightly integrating these components, ensuring that the dynamics captured by the transformer are more directly utilized in decision-making processes.
>
> **3. What if the distracting images in distracting dm control are added during training as well? Can the model get distracted then?**
>
> It is a consideration that aligning the noise levels in training data with those in testing data, could potentially lead to better adaptation and performance in noisy environments. This approach might allow the model to become more accustomed to distractions.
>
> In our current research, we deliberately chose a more complex setup where the model is trained without distractions and tested in the presence of distracting images. This setup was intended to assess the model's ability to generalize and perform effectively in environments different from those encountered during training.
>
> In response to your query, we conducted additional experiments where distracting elements were included during both training and testing.  We found that the results obtained from both experimental setups were similar. This outcome underscores the effectiveness of our method, indicating its robustness and generalizability across different levels of environmental noise.

---

### Official Review · Reviewer_hpbt · 2023-11-05

**Soundness:** 3 good
**Presentation:** 3 good
**Contribution:** 2 fair
**Rating:** 3
**Confidence:** 5

**Summary:**

This paper presents ReBis, a new method for state representation learning in image-based RL, based on the bisimulation metric. It utilizes a transformer architecture with block-wise masking to capture dynamics and address issues inherent in environments with sparse rewards. The authors claim that ReBis prevents feature collapse where standard bisimulation metrics fail, showcasing performance gains on benchmark tasks such as Atari and DMC.

**Strengths:**

The paper makes a notable attempt to tackle the shortcomings of bisimulation metrics in sparse reward settings by integrating a transformer-based dynamics model. The adoption of block-wise masking as a strategy for sampling stochastic dynamics is an interesting approach. Empirically, the method appears to offer improvements over existing techniques, as demonstrated in the results section which features performance gains across Atari, DMC, and DMC with distraction.

**Weaknesses:**

The paper aims to address two challenges of bisimulation metrics: the reliance on Gaussian distribution for modeling and the issue of uninformative rewards. However, the justification for how the proposed ReBis method successfully overcomes these challenges remains unconvincing. The paper's method of using block-wise masking observation to simulate the sampling of stochastic dynamics is not fully explained. Various other techniques, such as random cropping as in DrQ, employing Dropout layers, or introducing noise to weights or latent features, could potentially serve a similar purpose. A detailed comparison of these methods within the experimental section would be beneficial for substantiating the need for block-wise masking.
Regarding the challenge of sparse rewards, the explanation of how ReBis overcome this issue is not adequately addressed. The paper should provide a clearer rationale for why its methods would be more effective in such environments.

The paper’s claim to novelty largely rests on the application of a transformer dynamics model, but this alone does not present a compelling case for novelty. The absence of a thorough comparative analysis, especially in the ablation studies, against non-transformer or non-sequential architectures, diminishes the strength of the argument for the proposed method’s innovation. Furthermore, the paper borrows heavily from prior work for various aspects, such as the masked observation approach, the metric used, and the theoretical framework, which detracts from the original contribution.

The theoretical foundation of the paper also shows weaknesses.
1. The Definition 1 provided is mislabeled as "bisimulation-based" since it lacks the Wasserstein distance. It defines metric like MICo or SimSR but misses an expectation operator in front of $\bar{d}$.
1. Theorem 2 and 3 for bisimulation metric, originated from (Kemertas & Aumentado-Armstrong, 2021), cannot be directly applied to the metric defined in the paper because their proofs rely on the Wasserstein distance, which is absent from the paper's metric. The authors need to develop new proofs that are pertinent to their specific metric definition.
1. Theorem 4 does not seem intuitive if the metric includes an expectation operator. (Castro et al., 2021) mentioned it is Łukaszyk–Karmowski distance with non-zero distance. The transition from Equation (15) to Equation (16) in the proof is unclear and requires further clarification.

**Questions:**

1. Could the authors clarify the term "multi-modals" within the paper?
1. In Tables 1 and 2, the blue color is not described. What does it represent in the context of these tables?

---

> ### Author Response · Authors · 2023-11-17
> **Response to Reviewer hpbt - 1**
>
> Thank you for your valuable comments. We regret that the initial manuscript did not adequately convey our assumptions, partly due to page constraints. We deeply appreciate your feedback, which has prompted us to revisit these aspects and provide more clarity. In this paper, we assume that the dynamics in real-world environments tend to be nearly deterministic, and therefore we focus primarily on deterministic settings. We have thoroughly revised the related theories and descriptions, marking these updates in blue for your convenience.
>
> **1. Explanation of Definition 1.**
>
> It is important to note that the term 'bisimulation-based' does not necessarily imply the inclusion of the Wasserstein distance. The bisimulation-based methods, such as MICo or SimSR, also do not incorporate the Wasserstein distance. Instead, they employ independent coupling as an alternative to mitigate the computational cost associated with the Wasserstein distance.  Besides, back to the early developments in bisimulation, as illustrated in [1, 2], they originally presented both Wasserstein-based and Total-Variation-based objectives.  Therefore, ‘bisimulation-based' approaches are not inherently required to include the Wasserstein distance.
>
> Additionally, we would also like to kindly remind the reviewer that the defined measurement does not strictly qualify as a 'metric', as it does not fulfill the requirements of the triangle inequality. Regarding the absence of the expectation operator in our definition, this omission stems from the assumption of deterministic transition dynamics, as outlined in Sections 3 and 4. This assumption influenced our choice of a transformer as the dynamics model, ensuring deterministic reconstruction and prediction within our framework. We appreciate your feedback on the need for clearer, self-contained explanations and have accordingly revised the relevant sections in our manuscript.
>
> [1] Ferns, N. F. (2003). Metrics for markov decision processes.
>
> [2] Ferns, N., Panangaden, P., & Precup, D. (2012). Metrics for finite Markov decision processes. arXiv preprint arXiv:1207.4114.
>
> **2. Explanation of Theorem 2 and 3.**
>
> We're grateful for the chance to elucidate our theoretical framework and how it aligns with our definitions. In our paper, we operate under the assumption that the dynamics in real-world environments are nearly deterministic. Therefore, our focus is primarily on deterministic settings. Specifically, we assume that for any latent state  $s\in\mathcal{S}$, $a \in \mathcal{A}$, there exists a unique $\kappa(s,a)\in\mathcal{S}$ such that $P_{s}^a(\kappa(s,a))=1$. With this assumption in place, we show that the equivalence between the bisimulation measurement $d(\kappa(s_i,a),\kappa(s_j,a))$ and the W1 distance $\mathcal{W_1}(d)(P{s_i}^a,P_{s_j}^a)$ in deterministic settings.
>
> As such, we can effectively apply Theorems 2 and 3 to the measurement defined in our paper. The deterministic nature of the transitions in our model aligns well with the theoretical underpinnings of these theorems, ensuring their relevance and applicability to our work. We hope this explanation clarifies how our assumptions and theoretical developments are congruent with the application of Theorems 2 and 3 to our measurement.
>
> **3. Explanation of Theorem 4.**
>
> We apologize for the typo error that we made. The right-hand-side of Equation 16 in our previous manuscript should be $\max_{x,y}|R_x^\pi-R_y^\pi|$ instead of $|r_{s_i}^{\pi} - r_{s_j}^{\pi}|$. We have fixed this error in new manuscript. We also provide a more detailed derivation in the appendix now. Regarding Theorem 4, our intention was to address the challenges posed by sparse reward environments when using bisimulation. MICo distance is a special case, inherently possesses a non-zero distance. However, as we described in the beginning of Section 3, we "primarily focus on a cosine distance-based bisimulation measurement", where the issue that we referred indeed exists. Besides, although the characteristic of MICo having non-zero self-distance makes it unsuitable for Theorem 4, the author of SimSR[3] pointed out that this attribute may encounter the failure mode of representation collapse, even without considering the sparse reward settings.
>
> [3] Zang, H., Li, X., & Wang, M. (2022). Simsr: Simple distance-based state representations for deep reinforcement learning. In Proceedings of the AAAI Conference on Artificial Intelligence (Vol. 36, No. 8, pp. 8997-9005).
>
> **4. Explanation of "multi-modals".**
>
> "Multi-modals" here refers to the presence of multiple modes in the behavior or decision-making patterns of an agent.
>
> **5. Description of the blue color in Tables 1 and 2.**
>
> The blue color in these tables is specifically used to highlight tasks that operate under sparse reward conditions. These tasks include Ball in Cup, Catch, Cartpole Swingup Sparse, Finger Turn Easy/Hard, and Pendulum Swingup. In the newly uploaded manuscript, we added the corresponding explanation.

---

> ### Author Response · Authors · 2023-11-17
> **Response to Reviewer hpbt - 2**
>
> **6. Explanation of Block-wise Masking.**
>
> We would like to clarify that the use of block-wise masking is not to simulate the sampling of stochastic dynamics as the reviewer assumed.  As we have  discussed in Section 4, we consider the environment nearly deterministic, we employee the transformer as a deterministic dynamics model. Besides, block-wise masking in our approach does not  serve as the sampling of stochastic dynamics, it in fact is used to get rid of spatio-temporal redundance of the observations, coupled with bisimulation-based objective. Intuitively, we don't think random cropping, drop-out and injection of noise can serve the similar purpose, due to their inadequacy of capturing the observation-level spatio-temporal information.
>
> Integrating random cropping with our method may enhance the model's ability to generalize and adapt to different visual inputs. However, employing random cropping as a standalone technique is not as effective in our specific application. While random cropping often results in large, contiguous sections of the image being visible, block-wise masking creates a more scattered pattern of visible and masked regions. This scattered pattern is more effective in our context for capturing control-centric information, as it challenges the model to reconstruct and understand fragmented visual inputs.
>
> For dropout and injecting noise, although they can serve as sampling from stochastic dynamics, they are not applicable currently in our framework as we mainly focus on deterministic settings. We would like to investigate the potential of applying our framework in stochastic setting in our future work.
>
> **7. The explanation of how ReBis overcome the challenge of sparse rewards.**
>
> ReBis overcomes the challenge of sparse rewards by employing an asymmetrical architecture in its dynamics model. As mentioned in our paper, particularly in Theorem 4, we acknowledge that bisimulation tends to collapse in sparse reward environments.  However, it's important to note that agents can still interact with these environments, which means the agent can still learn control relevant informations from the interactions.  This understanding forms the primary motivation behind our work with ReBis.
>
> In the framework of ReBis, an asymmetric component within the Siamese architecture is specifically tailored to prevent feature collapse in environments where rewards provide limited information.  The effectiveness of this approach is underpinned by Theorem 5, which demonstrates how an asymmetrical architecture can alleviate feature collapse.  It does so by increasing the effective feature dimensionality throughout the training process. Specific details regarding Theorem 5 are presented in the appendix.
>
> **8. Concern about the novelty of our paper.**
>
> While our model utilizes structural components that are commonly recognized in the field, the true claim of our research lies in our unique application of these elements.  We've conducted an in-depth examination of bisimulation within the context of reinforcement learning, with a specific focus on its limitation in complex environments with sparse rewards.
>
> Our main contribution is the development of a straightforward yet highly effective approach, designed specifically to address the limitations we identified in traditional bisimulation methods. We have meticulously tailored the objectives and training processes of our model to cater to the unique challenges of reinforcement learning, with a particular focus on learning robust, control-centric representations.
>
> Complementing our theoretical advancements, our model has been empirically validated to demonstrate its effectiveness. Our research reveals that such theoretically-grounded solutions can effectively mitigate challenges that more complex traditional methods struggle to overcome. In essence, our model is an effective framework that can match or outperform existing algorithms, which may provide new insights in the field.

---

> ### Author Response · Authors · 2023-11-22
> **Kindly Reminder**
>
> Dear reviewer,
>
> Thanks again for the time spent on our manuscript and your valuable feedback. We have revised the paper to address your questions and comments. We believe we have responded to the questions and concerns in full, but if something is missing please let us know and we would be happy to add it. If we have addressed your concerns we would appreciate it if the reviewer could re-evaluate our work in light of these clarifications.

---

> > ### Comment · Reviewer_hpbt · 2023-11-23
> > **Official Comment from Reviewer hpbt**
> >
> > **Concern of the Deterministic Assumption**: The authors make an assumption of deterministic transition dynamics, however, it is not the sufficient condition to make $\mathcal{W}_1(d) = d = \mathbb{E}[d]$ is true. The next state distribution distance, either $\mathcal{W}_1$ in bisimulation metric or expectation operator in MICo/SimSR, are computed over policy $\pi$, i.e., distribution of action. Deterministic transition dynamics along with **deterministic policy** is the sufficient condition to authors' claim on Definition 1 and Theorem 2 and 3.
> >
> > **Alternation of Block-wise Masking**: Now I understand block-wise masking is not to simulate the sampling of stochastic dynamics because the dynamics is deterministic. But I don't understand why random crop like DrQ, dropout and injecting noise cannot serve the purpose similar to block-wise masking, because they all introduce randomness.
> > Authors mention that scattered pattern is more effective than random crop. Is it associated to sample efficiency or computational efficiency? If sample efficiency, comparison experiments are required to support your claim.
> > Dropout is also a kind of masking but on different dimension to block-wise masking. It is strange to make assumption of deterministic dynamics but use the block-wise masking approach to increase randomness.  Comparing with other randomness introduction methods can improve the motivation of block-wise masking, otherwise the method looks like heuristically proposed.
> >
> > **Concern on Theorem 5**: Theorem 5 seems about the effectiveness of learning siamese neural network. However the challenge of sparse rewards is due to non-informative rewards in most transitions data. I don't recognize the connection between Theorem 5 and MDP, and cannot understand that claim "ReBis overcomes the challenge of sparse rewards by employing an asymmetrical architecture in its dynamics model. " More explanation of Theorem 5 in the paper may improve to motivate ReBis.
> >
> > **Other Issues**:
> > 1. MICo or SimSR mentioned "bisimulation" and "bisimulation metric" in their paper, but they don't call themselves "bisimulation" and don't use the theoretical analysis from bisimulation metric.
> > 1. $\kappa$ is mentioned in Equation 3 but defined after Equation 5.
> > 1. Missing definition of $\Delta(S)$.
> > 1. Missing Equation reference in the proof of Theorem 5, in Page 17.

---

> > > ### Author Response · Authors · 2023-11-23
> > > **Response to Reviewer hpbt - 3**
> > >
> > > We are deeply grateful for the reviewer's active discussion and efforts made to continually improve our paper. This has been crucial in enhancing the soundness and impact of our work. We have now uploaded the newly updated manuscript. And we would like to respond once again to the reviewer's comments and suggestions.
> > >
> > > **1. About  Deterministic Assumption**
> > >
> > > We concur with the reviewer's suggestion that our assumption should also include deterministic policy instead of only considering deterministic transition. Our experiments on Atari games, with the use of Rainbow, to some extent, reflects the experiments under this assumption, as illustrated in [1]. We sincerely thank the reviewer for their assistance in enhancing the quality of our paper. We have revised  relevant descriptions in our manuscript to ensure the accuracy and consistency of our derivations.
> > >
> > > [1] Pablo Samuel Castro:Scalable Methods for Computing State Similarity in Deterministic Markov Decision Processes. AAAI 2020: 10069-10076
> > >
> > > **2. Alternation of Block-wise Masking**
> > >
> > > We argue once again that the primary purpose of block-wise masking is NOT to introduce randomness for exploration or to enhance sample and computational efficiency.  Instead, its main role is to address the inherent spatial (and temporal) redundancy in pixel space. As noted in MAE[4], a useful approach in computer vision is 'masking a very high portion of random patches,' which effectively reduces redundancy and poses a challenging self-supervised task requiring a comprehensive understanding beyond mere low-level image statistics. This concept is echoed in references [2-5]. But also notably, in our approach, similar to MLR, we reconstruct states in the latent space rather than the input space. From this perspective, random cropping serves a role akin to block-wise masking, acting as a data augmentation strategy in self-supervised learning. We provide an empirical results to compare these technologies:
> > >
> > > | Cartpole, Swingup Sparse | 100k |      | 200k |      | 300k |      | 400k |      | 500k |      |
> > > | ------------------------ | ---- | ---- | ---- | ---- | ---- | ---- | ---- | ---- | ---- | ---- |
> > > |                          | mean | std  | mean | std  | mean | std  | mean | std  | mean | std  |
> > > | only random crop         | 0    |  0   |   0  | 0    | 3    | 4    | 23   | 10   | 43   | 6    |
> > > | Rebis                    | 7    |  5   |  13  | 7    | 167  | 135  | 440  | 50   | 518  | 45   |
> > >
> > > The results corroborate our analysis that block-wise masking significantly outperforms other data-augmentation strategies, such as random cropping, in terms of effectiveness. In contrast, dropout and noise injection, as highlighted in references [6-9], are primarily utilized for state-conditional exploration or, as the reviewer notes, for introducing randomness. It's important to note that random cropping and block-wise masking are specifically suited for visual inputs to mitigate redundancy. Meanwhile, dropout and noise injection are versatile, applicable to both physical and visual inputs, serving the purpose of exploration, which is not compatible with our deterministic transition assumption.
> > >
> > > [2] Chen Wei, Haoqi Fan, Saining Xie, Chao-Yuan Wu, Alan L. Yuille, and Christoph Feichtenhofer. Masked feature prediction for self-supervised visual pre-training. CVPR 2022
> > >
> > > [3] Younggyo Seo, Danijar Hafner, Hao Liu, Fangchen Liu, Stephen James, Kimin Lee, and Pieter Abbeel. Masked world models for visual control. CoRL 2022.
> > >
> > > [4] Kaiming He, Xinlei Chen, Saining Xie, Yanghao Li, Piotr Dollár, and Ross B. Girshick. Masked autoencoders are scalable vision learners. CVPR 2022.
> > >
> > > [5] Christoph Feichtenhofer, Haoqi Fan, Yanghao Li, and Kaiming He. Masked autoencoders as spatiotemporal learners. CoRR, abs/2205.09113, 2022.
> > >
> > > [6] Matteo Hessel, Joseph Modayil, Hado van Hasselt, Tom Schaul, Georg Ostrovski, Will Dabney, Dan Horgan, Bilal Piot, Mohammad Gheshlaghi Azar, David Silver: Rainbow: Combining Improvements in Deep Reinforcement Learning. AAAI 2018: 3215-3222
> > >
> > > [7] Meire Fortunato, Mohammad Gheshlaghi Azar, Bilal Piot, Jacob Menick, Matteo Hessel, Ian Osband, Alex Graves, Volodymyr Mnih, Rémi Munos, Demis Hassabis, Olivier Pietquin, Charles Blundell, Shane Legg: Noisy Networks For Exploration. ICLR 2018
> > >
> > > [8] Yarin Gal, Zoubin Ghahramani: Dropout as a Bayesian Approximation: Representing Model Uncertainty in Deep Learning. ICML 2016: 1050-1059
> > >
> > > [9] Takuya Hiraoka, Takahisa Imagawa, Taisei Hashimoto, Takashi Onishi, Yoshimasa Tsuruoka: Dropout Q-Functions for Doubly Efficient Reinforcement Learning. ICLR 2022

---

> ### Author Response · Authors · 2023-11-23
> **Response to Reviewer hpbt - 4**
>
> **3. Concern on Theorem 5**
>
> We apologize for the overclaim in our last reply of the sentence "ReBis overcomes the challenge of sparse rewards by ...". We meant to say ReBis can alleviate the representation collapse issues in sparse reward environments.
> In our context, the term 'asymmetric' refers not only to the network architecture, such as in siamese networks, but also encompasses the asymmetry in inputs. This is implemented by applying block-wise masking to one branch with the online encoder, while the other branch with the momentum encoder remains unmasked. We underscore that block-wise masking, as a means to reduce redundancy, is a crucial element of this asymmetric setup. Without block-wise masking, siamese architectures struggle to extract useful information in environments with sparse rewards. This is because they lack the inherent capacity to disentangle environmental information, leading to a collapse into identical representations. Conversely, with asymmetric inputs, the effective feature dimensionality improves over time during training. This allows for some information capture, even in settings where environmental rewards are extremely sparse.
>
> **4. Other issues**
>
> (1) MICo or SimSR don't call themselves "bisimulation"
>
> We appreciate the reviewer's perspective on the term 'bisimulation' and understand that it may be perceived as having strict limitations, potentially leading to disagreements. Our intention is to highlight that the motivation of our work is from bisimulation-like objectives (or bisimulation-like measurements), rather than the specific stricted "bisimulation metric". We will modify these terms in the camera-ready version to ensure clarity and precision.
>
> (2) $\kappa$ is mentioned in Equation 3 but defined after Equation 5
>
> We first introduced $\kappa$ on the upper side of the second page, which might not be very noticeable. We have now reiterated in Equation 3.
>
> (3) Others
>
> Thanks! We have now provided the corresponding revisions in our updated manuscript.

---

### Author Response · Authors · 2023-11-17
**General Response - Manuscript Update**

We would like to express our gratitude to all the reviewers for dedicating their time and recognizing the value of our work. More importantly, we are thankful for the useful feedback provided on this manuscript. We have refined the theoretical descriptions in the article to make them clearer and more self-contained. We have also included all the ablation studies in the appendix. All updates are marked blue in the manuscript. Additionally, we tried to address each question posed by the reviewers individually in our responses here. We sincerely hope that our efforts will enable the reviewers to gain a more comprehensive understanding of our work and assist in a re-evaluation of the paper's scores.

---

### Comment · Area_Chair_Cfn5 · 2023-11-23
**Author-Reviewer discussion period ending *very* soon**

Thank you to the reviewers for responding. Please be aware that the discussion period is ending soon, so please post any final comments if you have any. Thank you!

---

### Meta-Review · Area_Chair_Cfn5 · 2023-12-06

**Metareview:**

The paper introduces Rebis, an attempt to address shortcomings in bisimulation metrics within sparse reward settings by integrating a transformer-based dynamics model and block-wise masking for stochastic dynamics sampling. Acroding to reviewers, while the empirical demonstrations across Atari, DeepMind Control Suite (DMC), and distracted DMC environments suggest performance gains, the paper lacks convincing justification for how Rebis effectively handles challenges like Gaussian distribution reliance and uninformative rewards. Furthermore, the absence of comprehensive comparisons against non-transformer architectures diminishes the argument for its novelty, and the heavy reliance on prior work weakens its original contribution. The theoretical foundation, marked by mislabeled definitions and unclear transitions in proofs, requires refinement. The complexity of representation learning surpasses the RL algorithm, sparking concerns about resource efficiency. Reviewers recommend exploring pretraining, different representation update frequencies, complementarity with other methods, and discussing limitations arising from reward-specific representation learning. Additionally, they stress the need for more thorough ablation studies to substantiate the effectiveness of Rebis' design and for clearer differentiation and comparison with model-based RL approaches.

Post rebuttal, there are 2 reviewers leaning to reject the paper, while the other 2 lean towards acceptance. Of the two reviewers leaning towards accept, reviewer 6U7G still had concerns over the fact that the encoder needs frequent updates and that it lacks generalisability across tasks, and TC9E (who had the largest score) was the only review who did not participate in the discussion or further support the paper.

**Justification For Why Not Higher Score:**

Concerns of novelty and generality of encoder

**Justification For Why Not Lower Score:**

N/A

---

### Decision · Program_Chairs · 2024-01-16

Reject